# Environmental heterogeneity can tip the population genetics of range expansions

**Matti Gralka[1†], Oskar Hallatschek[1,2]\***

[1]Department of Physics, University of California, Berkeley, Berkeley, United States;
[2]Department of Integrative Biology, University of California, Berkeley, Berkeley, United States

**Abstract** The population genetics of most range expansions is thought to be shaped by the competition between Darwinian selection and random genetic drift at the range margins. Here, we show that the evolutionary dynamics during range expansions is highly sensitive to additional fluctuations induced by environmental heterogeneities. Tracking mutant clones with a tunable fitness effect in bacterial colonies grown on randomly patterned surfaces we found that environmental heterogeneity can dramatically reduce the efficacy of selection. Time-lapse microscopy and computer simulations suggest that this effect arises generically from a local 'pinning' of the expansion front, whereby stretches of the front are slowed down on a length scale that depends on the structure of the environmental heterogeneity. This pinning focuses the range expansion into a small number of 'lucky' individuals with access to expansion paths, altering the neutral evolutionary dynamics and increasing the importance of chance relative to selection.
DOI: https://doi.org/10.7554/eLife.44359.001

**\*For correspondence:**
ohallats@berkeley.edu

**Present address:** [†]Department of Civil and Environmental Engineering, Massachusetts Institute of Technology, Cambridge, United States

**Competing interests:** The authors declare that no competing interests exist.

## Introduction

Stochasticity and its competition with deterministic forces plays an integral role in biology, such as in stochastic gene expression, cellular decision making, and cell differentiation (*Balázsi et al., 2011*). Stochastic processes are also at the heart of evolutionary dynamics: not only do the mutations entering a population occur at random times in random individuals and at random positions in their genome, but in addition the fate of a mutation and its clonal lineage is largely stochastic and only partly determined by its effect on the individual's fitness.

The random fluctuations in the frequency of a mutant allele due to the stochasticity associated with reproduction are called genetic drift. Genetic drift is particularly strong at the front of range expansions, where only a relatively small number of individuals at the front of the expansion contributes to future growth and thus has any influence on the future genotypic composition of the population. The neutral diversity and adaptation in spatially expanding populations has been studied in computer simulations (*Edmonds et al., 2004*; *Klopfstein et al., 2006*; *Kuhr et al., 2011*; *Kuhr and Stark, 2015*; *Lavrentovich and Nelson, 2014*; *Otwinowski and Krug, 2014*), in the field (*Ramachandran et al., 2005*; *White et al., 2013*; *Louppe et al., 2017*), and in microbial colonies (*Hallatschek et al., 2007*; *Fusco et al., 2016*; *Gralka et al., 2016b*; *Korolev et al., 2011*), which can serve as a useful model system because short generation times and ease of handling allow for quantitative investigations of the evolutionary dynamics of range expansions. In microbial colonies, nutrient gradients and mechanical effects limit the number of proliferating individuals to a small region close to the colony perimeter called the growth layer (*Grant et al., 2014*; *Gralka et al., 2016b*; *Warren et al., 2019*). For mutations occurring inside the growth layer, most mutant offspring are concentrated in a relatively small number of enormously successful lineages that manage to remain at the front and 'surf' on the expanding population wave (*Excoffier and Ray, 2008*). As a consequence, the evolutionary dynamics is different in microbial colonies compared with well-mixed

**eLife digest** Throughout evolution, countless populations have expanded their territories by invading new areas. This invasion can happen on the scale of kilometers and millennia – such as when humans migrated out of Africa – or millimeters and months, such as the growth of a population of cells in a solid tumor. During this expansion, mutations can occur that can either increase or decrease fitness in the new territory. If a favorable mutation occurs at the edge of the population, then it has plenty of room to expand. Such a mutation has a high chance of becoming established, and so it can have a very strong impact on the genetic makeup of the population. This increase in evolutionary advantage in individuals at the front is called "gene surfing".

This phenomenon is well known in populations living in "homogeneous" territories, where the new space a population is invading is more or less the same as the one they already occupy – think of the endless flat grasslands of the Siberian steppes. But in reality, many territories are not like that. What happens if the new territory is not completely homogeneous? For instance, if a species' expansion is impeded by a mountain range with a series of valleys.

Gralka and Hallatschek investigated how such changes in landscape could affect phenomena like gene surfing. Experiments using *E. coli* as a model system and computer simulations showed that a varied environment – such as roughness akin to a mountain range and valleys, but on a bacterial scale – had a strong influence on the fate of mutations arising in a population. Whether the environment is favorable for expansion or not in the place where the mutation happens becomes much more important than how the mutation itself affects fitness. So, if a beneficial mutation occurs at a cliff-edge, it is not likely to get far. But if it happens at a population edge by the bacterial equivalent of gently rolling hills, there is a much better chance of the mutation taking hold.

The findings suggest that the amount a population can adapt during expansion is limited, and it can even lead to the spread of harmful mutations in a population if they occur in just the right spot. Piecing together these scenarios is important in order to accurately infer the evolutionary history of a species based on mutations present in its genome now. This type of knowledge can also be useful in developing new treatments for cancers, making use of these evolutionary processes to slow or halt a tumor's expansion.

DOI: https://doi.org/10.7554/eLife.44359.002

populations. For instance, the clones (that is the collection of the initial mutant's offspring) of spontaneous neutral mutations often reach much larger sizes (*Fusco et al., 2016*), and existing beneficial variants can sweep to high frequency much faster in microbial colonies than in well-mixed populations (*Gralka et al., 2016b*). Conversely, deleterious mutations are predicted to remain at the population frontier for extended periods because genetic drift is strong at the front, which prevents selection from efficiently weeding out deleterious alleles (*Travis et al., 2007*; *Burton and Travis, 2008*; *Lavrentovich et al., 2016*; *Peischl et al., 2013*; *Gralka et al., 2016a*). The quantitative outcome of the competition of selection and genetic drift in microbial colonies is determined by the local shape and roughness of the front (*Gralka et al., 2016b*; *Farrell et al., 2017*), which in turn is determined by microscopic details, such as cell-cell adhesion or cell morphology shaping the mechanical interactions between cells (*Kayser et al., 2018b*; *Kayser et al., 2018a*; *Giometto et al., 2018*), although the direct mapping is typically unknown (*Farrell et al., 2017*).

The evolutionary effects of fluctuations at expanding microbial population fronts have been studied in depth, but these studies have focused only on fluctuations associated with the growth, division, and random motion of cells, whose strength may depend on *intrinsic* properties of the microbial species, in homogeneous environments. However, any range expansion will experience varying degrees of environmental heterogeneity, which can be viewed as a source of *extrinsic* noise. In macro-ecology, it has long been realized that environmental heterogeneity can dramatically alter the invasion dynamics of invasive species (*With, 1997*; *With, 2002*) or the genetic diversity after macroscopic range expansion (*Wegmann et al., 2006*). By contrast, how the evolutionary dynamics in microbial populations is affected by environmental heterogeneity in the form of, for example locally varying nutrient availability, temperature gradients, or imperfections in the growth substrate, has received much less attention. Experimental efforts have concentrated mostly on simple temporal

and spatial gradients in antibiotic concentration, which have been shown to facilitate the emergence of resistance in shaken cultures (*Lindsey et al., 2013*), microfluidic devices (*Zhang et al., 2011*) and on agar plates (*Baym et al., 2016*), as predicted by theory (*Greulich et al., 2012*; *Gralka et al., 2017*; *Hermsen et al., 2012*; *Hermsen, 2016*; *Hermsen and Hwa, 2010*).

The effects of spatial heterogeneity on evolutionary dynamics in expanding microbial populations have been studied in experiments only with neutral alleles in fixed geometries, such as isolated obstacles creating 'geometry-enhanced' genetic drift (*Möbius et al., 2015*; *Beller et al., 2018*). These studies have shown that obstacles obstructing locally the advance of the front can doom lineages that happen to lie on the blocked part of the expanding population, whereas unobstructed lineages close to the edge of the obstacles obtain a boost as they fill the vacant area behind the obstacle. Even for neutral alleles, however, not much is known about the evolutionary dynamics in more complex heterogeneous environments. Moreover, selection during range expansions can dramatically alter the population structure over just a few generations (*Gralka et al., 2016b*), but how the action of selection is affected by environmental heterogeneity has so far remained completely unexplored.

Here, we study the impact of complex environmental heterogeneities on the growth and evolutionary dynamics of microbial colonies. To this end, we introduce plasmid loss in *E. coli* as a model system for spontaneous mutations with tunable growth rate effects whose clones can be tracked under the microscope. By growing colonies on solid substrates with a weakly patterned surface with random microscopic features much bigger than individual cells, but much smaller than the whole colony, we find that environmental heterogeneity can overpower selection such that even strongly beneficial mutations are unable to establish at rates higher than expected for neutral mutations. Using a minimal computer model of populations expanding in randomly disordered environments, we show that dramatic changes in the efficacy of selection can arise from a local 'pinning' of the expansion front, whereby stretches of the front are slowed down on a length scale that depends on the structure of the environmental heterogeneity. This pinning focuses the range expansion into a small number of individuals with access to expansion paths, increasing the importance of chance and thus limiting the efficacy of selection. We expect these results to generalize to other spatially growing populations, such as biofilms, tumors, and invasive species, when the growing population front is transiently hindered by the local environment.

## Results

### Experiments

We grew colonies from single cells of a strain of *E. coli* MG1655 carrying a plasmid that is costly to produce, resulting in a $20\%$ growth rate disadvantage $s$ in plasmid-bearing cells compared to their plasmid-less (but otherwise isogenic) conspecifics (*Figure 1*). This strain loses the plasmid stochastically at a rate of about $5 \times 10^{-3}$ per cell division (approximately independent of antibiotic concentration, see *Figure 1—figure supplement 1*). The plasmid codes for a fluorescence gene and confers resistance to the antibiotic doxycycline (*dox*, a tetracycline analog) such that when grown at increasing antibiotic concentrations, cells missing the plasmid grow more and more slowly, until they eventually grow more slowly than cells harboring the plasmid (for $[dox] > 0.3 \mu g/\mathrm{ml}$), despite the inherent cost associated with bearing the plasmid. This allowed us to treat plasmid loss effectively as a spontaneous mutation whose fitness effect $s$, that is the relative growth rate difference between the plasmid-bearing ('wild type') and non-bearing ('mutant') cells, we can finely tune from +20% to -15% by varying the amount of doxycycline in the growth media (see *Figure 2—figure supplement 1*), thus making the mutation either beneficial, neutral or deleterious. Since plasmid loss in our system is also coupled with a loss in fluorescence, we can easily detect mutant clones, that is the individual cell that incurred the mutation originally and its offspring, under the microscope as dark patches in the colonies (see *Figure 1*), allowing us to observe the evolutionary dynamics in real time. Our approach extends previous experimental model systems for evolutionary dynamics during microbial range expansion that employed either an initial mixture of wild-type and mutant cells (*Hallatschek et al., 2007*; *Van Dyken et al., 2013*; *Müller et al., 2014*; *Gralka et al., 2016b*; *Korolev et al., 2012*; *Kayser et al., 2018a*) or were confined to spontaneous neutral (*Fusco et al., 2016*) or deleterious (*Lavrentovich et al., 2016*) mutations. The ability to track spontaneous mutations in colonies grown

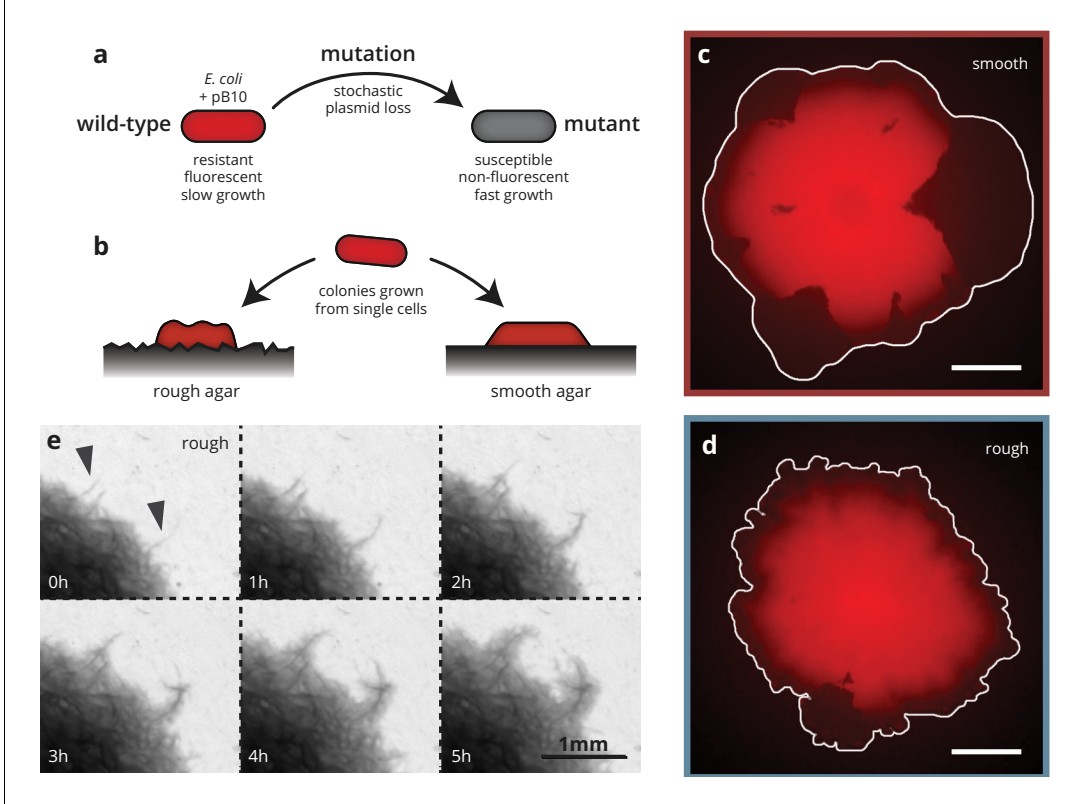

**Figure 1.** Substrate roughness impacts bacterial colony morphology. (**a** and **b**) Colonies of *E. coli* were grown from single cells harboring a plasmid containing a fluorescence gene and a resistance cassette. Loss of the plasmid leads to non-fluorescent ('mutant') sectors in the population that expand at the expense of the fluorescent ('wild-type') population. Colony morphology depends on whether the colony is grown on smooth (**c**) or rough (**d**) agar surfaces (Materials and methods). White scale bars are 2 mm. (**e**) On rough surfaces, troughs in the surface direct growth along them, leading to locally accelerated regions (arrows) that widen slowly and are eventually integrated with the bulk of the population.

DOI: https://doi.org/10.7554/eLife.44359.003

The following source data and figure supplements are available for figure 1:

**Figure supplement 1.** Likelihood of mutation rate for two different concentrations of doxycycline.

DOI: https://doi.org/10.7554/eLife.44359.004

**Figure supplement 1—source data 1.** Source data for *Figure 1—figure supplement 1* (Likelihood of mutation rates).

DOI: https://doi.org/10.7554/eLife.44359.005

**Figure supplement 2.** Analysis of substrate roughness.

DOI: https://doi.org/10.7554/eLife.44359.006

**Figure supplement 2—source data 1.** Source data for *Figure 1—figure supplement 2* (Five stylus runs to measure substrate roughness).

DOI: https://doi.org/10.7554/eLife.44359.007

**Figure supplement 3.** Roughness of *E.coli* colonies on smooth and rough substrates, defined as the mean square deviation from the best-fit circle.

DOI: https://doi.org/10.7554/eLife.44359.008

**Figure supplement 3—source data 1.** Source data for *Figure 1—figure supplement 3* (Roughness of smooth and rough colonies).

DOI: https://doi.org/10.7554/eLife.44359.009

from single cells is essential to ensure identical starting conditions in our experiments, allowing a quantitative comparison of the evolutionary outcomes between the two growth conditions.

To investigate the impact of environmental heterogeneity on colony growth and adaptation dynamics, we grew the colonies on two different substrates (*Figure 1*): standard, 'smooth', agar plates as well as agar surfaces with random microscopic features, created by depositing filter paper onto melted agar and removing it after cooling and drying (see Materials and methods). The resulting substrate had an average roughness (i.e., standard deviation of the substrate height) of 10 μm with ridges and valleys much wider than individual cells (about 15–30 μm), but small compared to whole colonies (see *Figure 1—figure supplement 2* for a detailed characterization of the rough

substrates). Notably, the width of the valleys is comparable to the growth layer width of *E. coli* colonies (*Gralka et al., 2016b*), which is the fundamental length scale characterizing the range of mechanical interactions within microbial colonies (*Kayser et al., 2018a*).

Colonies grew more slowly on these rough substrates compared to smooth substrates (*Figure 2a*), but this disorder-induced reduction in radial growth rate was consistent between wild-type and mutant cells, such that their selective difference $s$, defined as the difference between colony expansion rates, normalized by the wild-type colony expansion rate, was independent of surface structure (*Figure 2b–c*). Colonies grown on rough substrates (hereafter called 'rough' colonies) also had a rougher front line (see *Figure 1d* and *Figure 1—figure supplement 3*) than those grown on smooth substrates ('smooth' colonies), and displayed branch-like outgrowths where the bacteria tended to colonize grooves in the agar surface much faster than the surrounding areas (*Figure 1e*, arrows). These branches grew far ahead of the rest of the population, becoming visible at a width of about 20 µm (consistent with the width of the valleys in the substrate), and broadened as they were incorporated into the bulk of the colony. This kind of growth pattern is reminiscent of the 'pinning' phenomenon observed in the study of interfaces in heterogeneous media, such as the capillary rise of water or autocatalytic fluid interfaces in porous media (*Delker et al., 1996*; *Atis et al., 2015*), macro-ecological species invasions (*Keitt et al., 2001*), or magnetic domains (*Lemerle et al., 1998*). Pinning refers to the effect whereby certain regions of an expanding interface are slowed or even stopped entirely in their advance by heterogeneities, whereas other regions can advance unimpeded. Given the importance of the front morphology for evolutionary dynamics (*Gralka et al., 2016b*; *Farrell et al., 2017*), we hypothesized that by changing the growth patterns of rough colonies, the structured agar surface should also impact the dynamics of spontaneous mutations.

The primary readout of our experiments is the final mutant frequency $f_{MT}$ in the colony and the number of surviving mutant clones (sectors) as a function of the selective advantage $s$ that the mutation confers. These measures are proxies for the degree of adaptation of the population during the expansion process and the success probability of spontaneous mutations in shaping the composition of the population, respectively. Thus, our system gives us direct access to population genetic measures of interest. Alternatively, one can measure the frequency of mutants at the colony perimeter, which has a more direct influence on the future genetic composition of the population at the front. However, this measure can quickly become uninformative for beneficial mutations as mutants overtake the whole perimeter, and it is often difficult to measure accurately because of the low fluorescence signal at the front. The mutant frequency averaged over the whole population still gives a good, if conservative, measure of the mutant frequency at the perimeter.

On smooth substrates, in accord with previous experimental results (*Gralka et al., 2016b*; *Korolev et al., 2012*), advantageous mutants increased in frequency $f_{MT}$ rapidly as the colony grew (see *Figure 2—figure supplement 2* for an analysis of the mutant dynamics over time): for $s = 0.2$, mutants made up roughly half of the total population (*Figure 2d*) after 72 hrs. At higher antibiotic concentration, mutants became first neutral and eventually deleterious (*Figure 2—figure supplement 1*), and accordingly, the final mutant frequency was lower, decreasing approximately exponentially with the fitness cost $s$ (*Figure 2—figure supplement 4*) such that strongly deleterious mutants made up only a small fraction of the final population.

On rough substrates, deleterious and neutral mutants remained at frequencies comparable to those observed in smooth colonies at the corresponding values of the fitness (dis)advantage $s$. However, in contrast to smooth colonies, beneficial mutants in rough colonies did not increase in frequency with $s$ relative to the neutral case. This finding is surprising, given that the growth rate advantages of mutant over wild-type colonies were the same on rough and smooth substrates (*Figure 2c*). Thus, this apparent inefficacy of selection in affecting evolutionary outcomes was not caused by an altogether elimination of growth rate differences. Instead, a closer look at the colony growth dynamics on rough substrates, shown in *Figure 1e*, suggests a different mechanism: the surface structure of the rough substrate constrains and guides growth along predetermined paths where growth proceeds faster than in the immediate surroundings, such that any mutation can only be successful, that is establish a sector and thus rise to high frequency in the population, if it happens to arise in one of the branch-like regions of accelerated growth. Conversely, a beneficial mutant clone will be unable expand even if its per capita growth rate is higher than its wild-type neighbors if the mutation occurs in a portion of the front that is slowed down by the environmental heterogeneity.

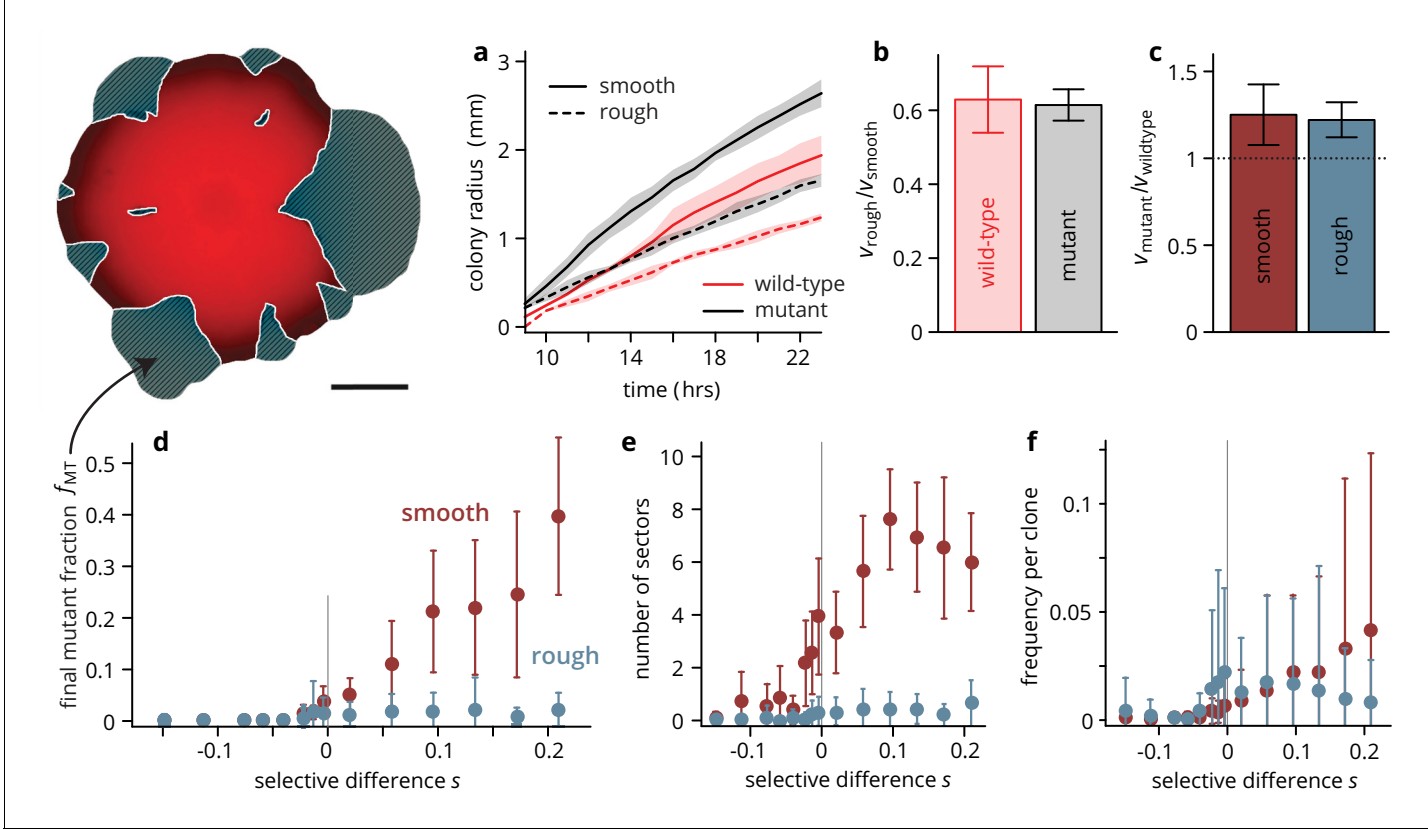

**Figure 2.** Substrate roughness limits the efficacy of selection in bacterial colonies. (a–c) The roughness of the growth substrate reduces the radial expansion rate but has no impact on the relative growth rates between mutants and wild type (averages over four colonies per condition, shaded areas in panel (a) are ± 1 standard deviation). (d) The final mutant frequency $f_{MT}$ (measured as the area fraction of non-fluorescent clones in the colonies, shown in white outline in sketch above) after 3 days of growth for mutants of a given fitness (dis)advantage $s$ differed strongly depending on the roughness of the surface they were grown on, despite relative growth rate differences $s$ being comparable (c) (throughout, points are averages over colonies, see *Table 1*, error bars are ± 1 standard deviation). The number of establishing sectors (e) was much smaller in rough colonies, with typically no more than two sectors per colony even for large $s$. The sizes of individual clones are broadly distributed on both substrates (see also *Figure 3*), but the mean clone size (f) did not vary as strongly with $s$ in rough colonies as it did in smooth colonies.

DOI: https://doi.org/10.7554/eLife.44359.010

The following source data and figure supplements are available for figure 2:

**Source data 1.** Source data for *Figure 2*.
DOI: https://doi.org/10.7554/eLife.44359.020
**Figure supplement 1.** Selective difference between mutants and wild type at different concentrations of doxycycline.
DOI: https://doi.org/10.7554/eLife.44359.011
**Figure supplement 1—source data 1.** Source data for *Figure 2—figure supplement 1* (Fitness measurements at various concentrations of doxycycline).
DOI: https://doi.org/10.7554/eLife.44359.012
**Figure supplement 2.** Mutant frequency $f_{MT}$ as a function of radius $r$ (see panel (c) for schematic), for smooth (a–b) and rough (d–e) colonies.
DOI: https://doi.org/10.7554/eLife.44359.013
**Figure supplement 2—source data 1.** Source data for *Figure 2—figure supplement 2* (Mutant frequency $f_{MT}$ as a function of radius $r$ for smooth and rough colonies).
DOI: https://doi.org/10.7554/eLife.44359.014
**Figure supplement 3.** Population size of colonies estimated by measuring colony height and area, and by resuspending and measuring the resulting optical density.
DOI: https://doi.org/10.7554/eLife.44359.015
**Figure supplement 3—source data 1.** Source data for *Figure 2—figure supplement 3* (Colony sizes on rough and smooth substrates).
DOI: https://doi.org/10.7554/eLife.44359.016
**Figure supplement 4.** Final frequency of mutants after 3 days of growth, on smooth (red) and rough (blue) plates.
DOI: https://doi.org/10.7554/eLife.44359.017

*Figure 2 continued*

**Figure supplement 5.** Mutant frequency measured in colonies grown on smooth and rough substrates at five doxycycline concentrations (converted to fitness differences $s$ as described in the main text) in a biological replicate experiment.
DOI: https://doi.org/10.7554/eLife.44359.018

**Figure supplement 5—source data 1.** Source data for *Figure 2—figure supplement 5* (Mutant frequency $f_{MT}$ vs selective difference $s$ in replicate experiment).
DOI: https://doi.org/10.7554/eLife.44359.019

If this proposed mechanism is indeed the root cause for the apparent inefficacy of selection on rough substrates, then one would expect the number of successful mutants, that is those that manage to establish sectors, to be independent of the mutant's selective advantage $s$. Indeed, this is what we observed: the probability of forming a sector, which quantifies the evolutionary success probability of individual mutations, increased with $s$ in smooth colonies (*Figure 2e*), but was constant in rough colonies as long as $s > 0$. Notably, the establishment probability was extremely low in both scenarios: Even for the most advantageous mutants in smooth colonies, we estimate $u \sim 10^{-7}$ per mutation making evolutionary success an extremely rare event. The low success probability is a consequence of two processes: firstly, the mutation must occur in a favorable location, namely in the first layer of cells at the front of the population (*Gralka et al., 2016b*), which reduces the number of mutations eligible for sector formation by a factor of about 1000 (assuming a growth layer width of 10 cells and a colony height of about 100 cells, see *Figure 2—figure supplement 3*). We estimate that about 2000 mutations per colony arose in favorable positions, each of which had an establishment probability of order $10^{-3}$. Secondly, each eligible mutation has to survive genetic drift, which in microbial colonies is manifest in the random fluctuations in the sector boundaries as a consequence of stochastic cell growth and division, and subsequent cell motion due to mechanical pushing of cells on each other (*Hallatschek et al., 2007*; *Farrell et al., 2017*; *Kayser et al., 2018a*).

The low establishment probability means that most mutations will not manage to create sectors, but instead they will form so-called bubbles, that is individual mutant clones that have lost contact with the front. We extracted the size of mutant clones, both bubbles and sectors, by measuring the individual areas of non-fluorescent patches in the colony micrographs. The resulting clone size distribution $P(X > x)$ (clone areas normalized by colony area, shown in *Figure 3*) is related to the site frequency spectrum in population genetics, where it can be used to predict rare evolutionary outcomes such as fitness valley crossing (*Weissman et al., 2009*) and evolutionary rescue (*Fusco et al., 2016*), and is well understood for toy models of microbial colonies (*Fusco et al., 2016*; *Otwinowski and Krug, 2014*). For neutral mutations, the clone size distribution is expected to be broad up to a shoulder indicating the typical size of the largest expected bubble. In our experiments, we indeed observed a broad shoulder-like distribution for neutral mutations, consistent with earlier experiments using population sequencing (*Fusco et al., 2016*). In smooth colonies, beneficial mutations created a larger number of bulging sectors, leading to an even broader distribution with maximum clone sizes of almost half the population, while the distribution for strongly deleterious mutations was cut off at small clone sizes. This clone size distribution is consistent with our initial observation that a larger selective advantage $s$ gave rise to a larger overall mutant frequency, but it also shows that even at the largest $s \approx 0.2$, most mutant clones remained small, with more than half of the visible clones reaching frequencies of at most 1%. By contrast, the clone size distributions obtained from rough colonies were virtually indistinguishable for all $s > 0$, whereas we observed the same cut-off for large clones for deleterious mutations.

In summary, a microscopically randomly patterned growth surface had several effects on the population and evolutionary dynamics of our colonies. The heterogeneity of the substrate decreased the radial growth rate during early colony growth and gave rise to colonies with an overall rougher morphology. In terms of evolutionary dynamics, heterogeneity decreased the dependence of the final mutant frequency (or, equivalently, the rate of adaptation) on the selective effect of mutations. These effects are large, even though the perturbation we impose seems relatively weak. After all, the rough substrate is only distinguished from the smooth substrate by troughs and elevations of order ten micrometers, and colony growth rate differences are consistent for both substrate types (*Figure 2c*).

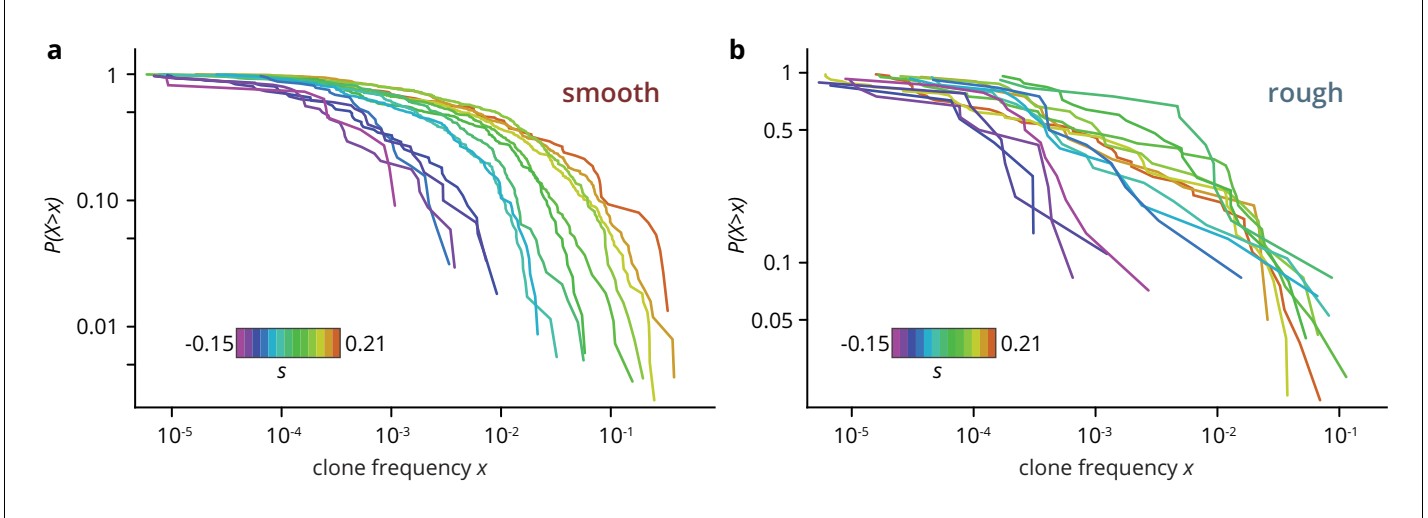

**Figure 3.** Clone size distribution $P(X>x)$ for colonies grown on smooth (**a**) and rough (**b**) substrates. Deleterious mutants are shown in magenta tones; their clones are typically small. Neutral clones are shown in green; their size distribution had a broad tail. Advantageous mutations in smooth colonies were even more broadly distributed, as large sectors establish more often. By contrast, beneficial clones in rough colonies had size distributions that were indistinguishable from the distribution for neutral mutations.

DOI: https://doi.org/10.7554/eLife.44359.021

The following source data is available for figure 3:

**Source data 1.** Source data for *Figure 3*.

DOI: https://doi.org/10.7554/eLife.44359.022

## Minimal model

How can such a relatively small change in environmental conditions have such a dramatic effect on the evolutionary dynamics? Above, we have proposed that the transient colony pinning seen in *Figure 1e* may be responsible, by giving a boost to certain regions irrespective of whether that region harbors beneficial mutants or not. However, the growth of the colony and the mutational dynamics within it are highly complex processes affected by the mechanical properties of growing cells and their interactions with each other and the growth substrate (*Grant et al., 2014*; *Boyer et al., 2011*; *Kayser et al., 2018a*; *Giometto et al., 2018*; *Farrell et al., 2017*). In addition, the substrate heterogeneity in our experiments is complex and characterized by long-range correlations (see *Figure 1—figure supplement 2*). This raises the question whether our key findings may hinge on these complexities, or whether much simpler uncorrelated heterogeneities in growth rates can also have comparable effects on population genetics.

To answer this question, we have devised a minimal simulation model for populations expanding in heterogeneous environments. Briefly, colonies grow from single cells on a square lattice, only cells with empty neighbors can divide, and a wild type can mutate upon cell division with probability $\mu$ to the mutant type carrying a fitness advantage or disadvantage $s$ (Materials and methods). Disorder sites (density $\rho$) confer a reduced growth rate $k$ ($0 \leq k < 1$) to any individual growing on it. We call $k$ the transparency of the disorder sites; for $k = 0$, we refer to the disorder sites as (impassable) obstacles. The simplicity of the model allows us to explore exhaustively the whole parameter space in $k$ and $\rho$. Our model is based on the classical Eden lattice model (*Eden, 1961*) that is commonly used to model growing microbial colonies (*Gralka et al., 2016b*; *Fusco et al., 2016*; *Ben-Zion et al., 2019*). The Eden model without environmental heterogeneity is in the so-called KPZ universality class, that is its statistical properties are described the KPZ equation, a classical model of interface growth (*Kardar et al., 1986*). The KPZ equation has been extended to include environmental heterogeneity (discussed in detail in the Materials and methods) which was shown to induce a pinning transition: the environmental heterogeneity induces a characteristic length scale on which the interface cannot advance (is *pinned* by the heterogeneities). Thus, adding environmental heterogeneity to the Eden model may likewise introduce a pinning transition under certain conditions,

making our generalized Eden model a natural candidate for a minimal model of range expansions in heterogeneous environments.

Indeed, in agreement with our experiments, increasing the density $\rho$ of disorder sites leads to a decrease in the radial colony expansion speed in our simulations (*Figure 4b*) that becomes more extreme as the obstacle transparency $k$ goes to 0. For small $k \ll 1$, the expansion speed decreases first slowly and then rapidly as the density reaches a critical value $\rho_c \approx 0.4$. For impassable obstacles ($k = 0$) at densities $\rho > \rho_c$, the obstacles form a closed ring around the incipient colony and prevented further growth (*Figure 4b*, black line). This is the anticipated pinning transition, which occurs in our model at a critical density $\rho_c \approx 0.4$. This critical density corresponds to the scenario where the colony can no longer percolate through the network of obstacles, suggesting that $\rho_c$ is equivalent to the site percolation threshold $1 - 0.592 \ldots = 0.407 \ldots$ (*Bunde et al., 1985*; *Barabási and Stanley, 1995*). Notably, while the percolation transition only occurs at the critical point $k = 0$, non-zero but small values of $k$ give rise to transient pinning near the critical density that is essentially indistinguishable

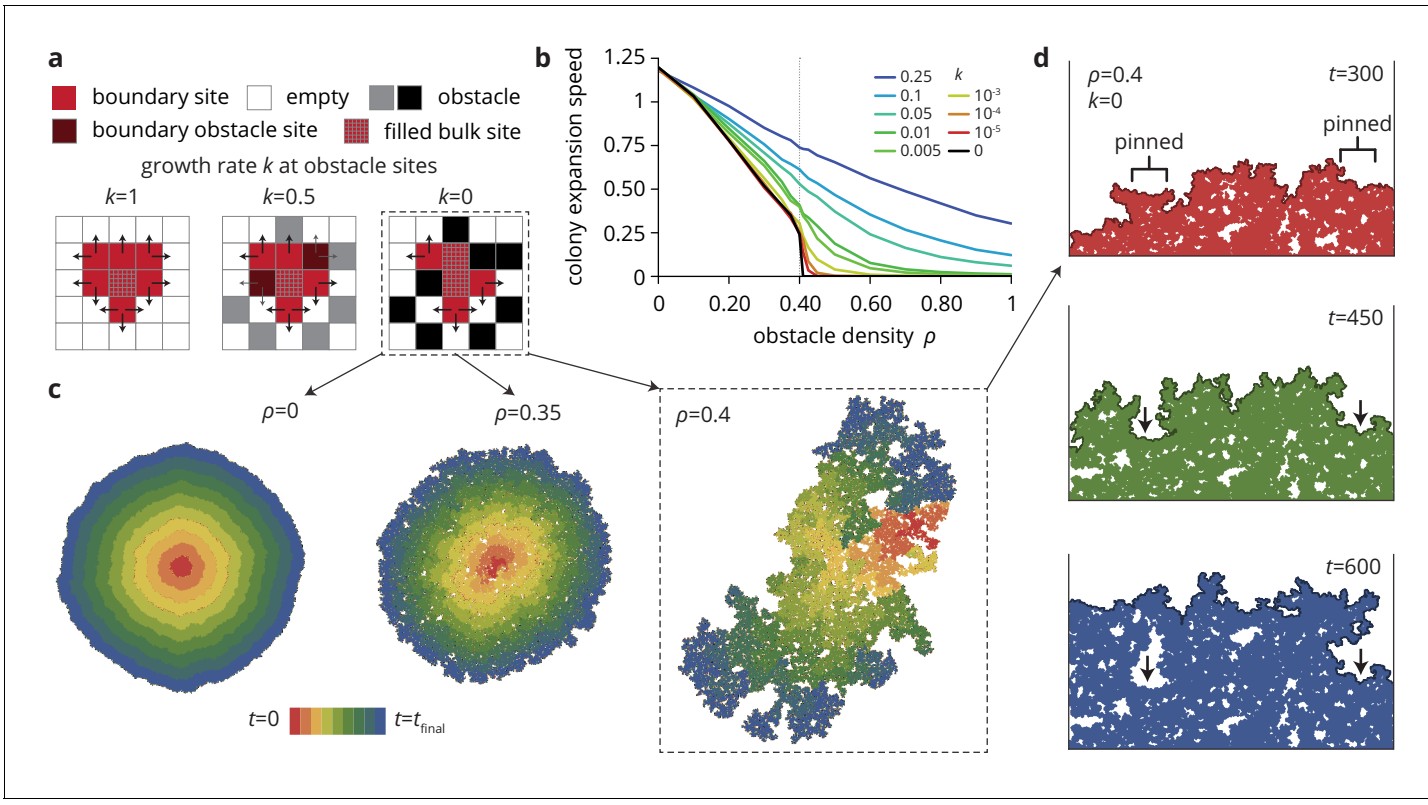

**Figure 4.** A minimal model captures the change in colony morphology in heterogeneous environments. (a) The simulation proceeds by allowing cells with empty neighbors ('boundary sites') to divide into empty space. Disorder sites (gray/black, density $\rho$) are randomly distributed over the lattice. (b) As the obstacle transparency, that is the growth rate $k$ at disorder sites, is reduced, the colony expansion speed decreases as a function of the obstacle density. For impassable disorder sites ('obstacles', $k = 0$), the expansion speed vanishes above a critical obstacle density $\rho_c \approx 0.4$, beyond which the population was trapped and could not grow to full size. (c) As $\rho$ approached $\rho_c$ colony morphology became increasing fractal. Colors from red to blue show the colony shape at different times. (d) Near the critical obstacle density $\rho_c \approx 0.4$, only few individual sites near the front have empty neighbors, leading to local *pinning* of the front (arrows), where the front can only progress by lateral growth into the pinned region.

DOI: https://doi.org/10.7554/eLife.44359.024

The following source data and figure supplements are available for figure 4:

**Source data 1.** Source data for *Figure 4* (Colony expansion rate in heterogeneous environments).
DOI: https://doi.org/10.7554/eLife.44359.027

**Figure supplement 1.** Characterization of Eden clusters with obstacles ($k = 0$).
DOI: https://doi.org/10.7554/eLife.44359.025

**Figure supplement 1—source data 1.** Source data for *Figure 4—figure supplement 1* (Analysis of colony expansion rate (v(rho).xslx), interface width as a function of window length (w(l).xslx) and time (w(t).xslx), and saturation width (Wsat.xslx)).
DOI: https://doi.org/10.7554/eLife.44359.026

from the $k = 0$ case over time scales shorter than $1/k$, while still allowing the expansion to progress indefinitely, albeit slowly. Close to the pinning transition, small changes in obstacle density can have dramatic effects: not only does the colony expansion speed decrease rapidly, but the colony morphology changes drastically and, as we show below, so do the evolutionary dynamics.

As a result of the local pinning of the colony interface, the colony morphology depends on the density of obstacles (*Figure 4c*), most drastically for impassable obstacles ($k = 0$) on which we concentrate for now. Without obstacles, the colonies are compact and relatively smooth. At intermediate obstacle densities (*Figure 4c*, middle), colonies are punctured by small holes and the overall density of the colony decreases. At the critical density $\rho_c$ the colony is characterized by the fragmented morphology of percolation clusters with a large number of holes and a very rough exterior (see *Figure 4—figure supplement 1* and Materials and methods for a quantitative analysis of the colony interfaces). Below the critical obstacle density, the interface is pinned locally over a length scale that depends on the proximity to the pinning transition; the whole interface becomes pinned when this length scale reaches the system size, whereas the interface is unaffected on length scales much larger than this pinning length (see *Theory*). An example of local pinning is shown in *Figure 4d*, where the interface can only advance when the individuals located in unpinned portions of the front grow into the pinned areas (indicated by arrows). This process is equivalent to the branch-like outgrowths in the experiments (*Figure 1e*) which correspond to unpinned portions of the front outgrowing the pinned areas. In the following, we show how the changes in colony morphology produced by the environmental heterogeneity affect the evolutionary dynamics.

We begin by replicating the experimental situation to assess the efficacy of selection in the presence of environmental heterogeneity. We simulated mutations conferring a selective advantage $s$ (that is increasing the growth rate by a factor $1 + s$), shown in *Figure 5*. Transparent obstacles ($k = 0.1$, *Figure 5a*) only have a relatively mild effect on the mutant frequency $f_{MT}$. For any value of the obstacle density $\rho$, $f_{MT}$ increases roughly exponentially with $s$, but the dependence on obstacle density is non-monotonic. This is easiest to see for the most beneficial mutations: the final mutant frequency $f_{MT}$ at $s = 0.2$ is lower at intermediate $\rho \approx 0.5$ than at the extremes $\rho \approx 0$ or $\rho \approx 1$. Intuitively, this non-monotonicity results from a symmetry between high and low $\rho$: in both cases, there is only a small fraction of sites of the 'other' type (i.e., disorder sites at low $\rho$ or regular sites at high $\rho$), and their density is too small to effect a strong change in the population genetics. The reduction in the sensitivity of $f_{MT}$ to $s$ at intermediate $\rho$ becomes much more dramatic as the obstacle transparency $k$ is decreased (see *Figure 5b* for the extreme case $k = 0$). A similar pattern is found for the final frequency of neutral mutants ($s = 0$), which is largest at intermediate $\rho$.

To quantify the effects of varying $k$ and $\rho$ and summarize the simulation results over many parameter combinations, we introduce the selection efficacy $k_s$ and the neutral diversity $f_0$ by parametrizing the mutant frequency with an exponential function $f_{MT}(s) = f_0 e^{k_s s}$. Although this choice is merely a heuristic, rescaling the mutant frequency curves for a range of values of $\rho$ and $k$ by the fitted values for the neutral diversity $f_0$ and the selection efficacy $k_s$, all points fall close to a single master curve given by a simple exponential (*Figure 5f*).

The selection efficacy $k_s$ and the neutral diversity $f_0$ have a minimum and maximum near $\rho_c$, respectively, which is increasingly sharp as $k$ approaches the critical point $k = 0$. At the critical point, the selection efficacy vanishes entirely as $\rho$ approaches $\rho_c$ (*Figure 5c,d*). Thus, selection is completely unable to affect the final mutation frequency as the critical point is approached. The virtual independence of the evolutionary dynamics of the per-capita fitness $s$ holds even at the scale of individual clones, whose size distributions for different values of $s$ are practically indistinguishable for obstacle densities near $\rho_c$ (*Figure 5—figure supplement 1*). Importantly, while we find a proper phase transition in the evolutionary dynamics only at the critical point ($k = 0$), the percolation transition is also manifest in populations grown in generic heterogeneous environments with $k > 0$ which do not give rise to a percolation transition. As a consequence, tiny changes in environmental parameters near a non-trivial critical obstacle density $\rho_c$ can have a dramatic effect on the population growth dynamics and colony morphology as well as its evolutionary dynamics. The close connection between colony morphology and evolutionary dynamics is underscored by the empirical observation that the two descriptive parameters, the selection efficacy $k_s$ and the neutral diversity $f_0$, introduced as independent parameters measured directly from the simulations, are not independent in practice (*Figure 5e*). Plotting $k_s$ vs. $f_0$ for various choices of $k$ and $\rho$ reveals that the two parameters represent

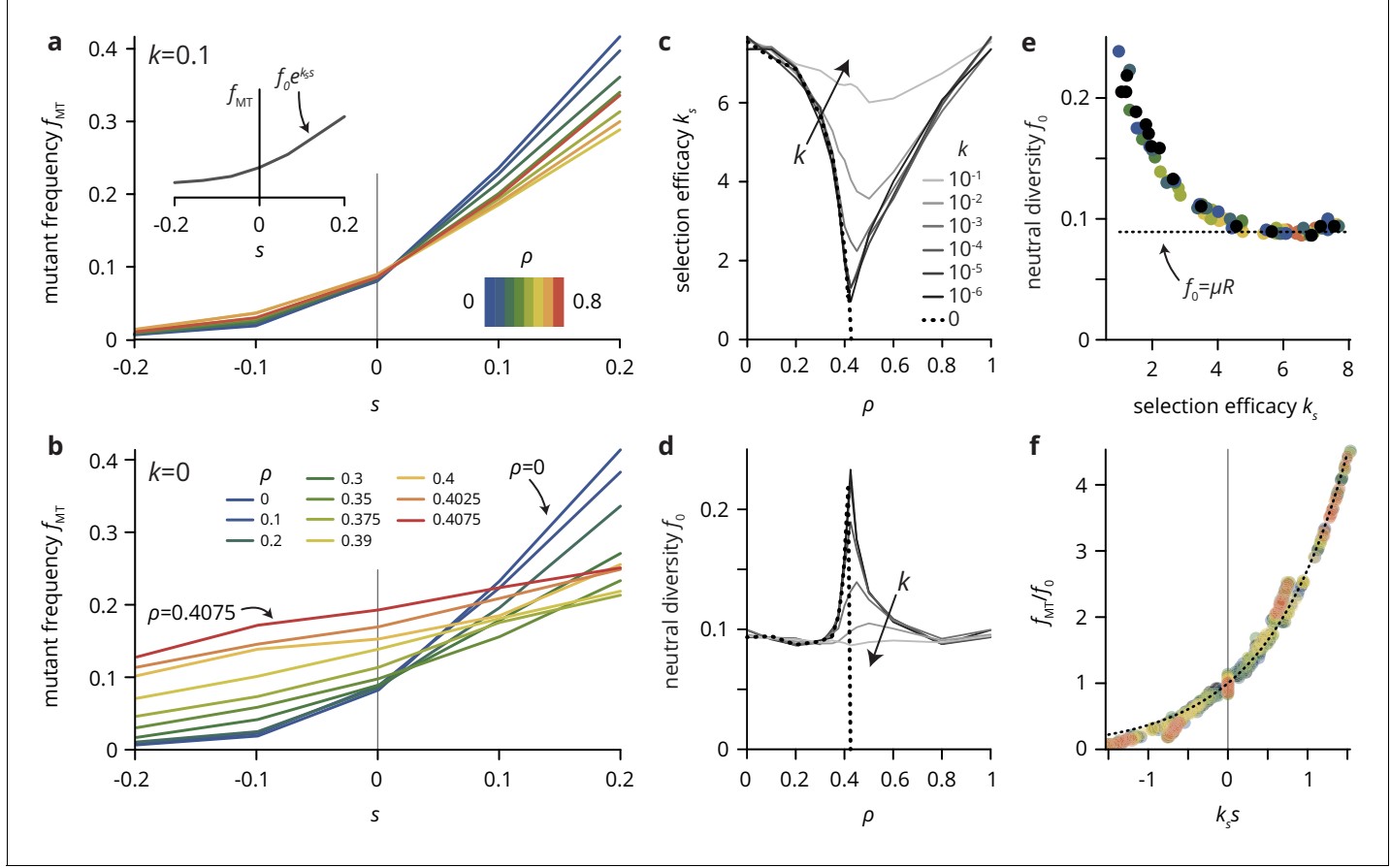

**Figure 5.** A simple lattice model reproduces strong impact of environmental heterogeneity on evolutionary dynamics. (a,b) The mutant frequency $f_{MT}$ as a function of the selective advantage $s$ of the mutants always increases with $s$ to a degree that depends on the density $\rho$ and the transparency $k$ of the disorder sites. (c,d) Parametrizing the mutant frequency for various values of $k$ and $\rho$ heuristically with $f_{MT} = f_0 e^{k_s s}$ (panel a, inset), we fit the neutral diversity $f_0$ (shown in panel d) and the selection efficacy $k_s$ (panel c). The selection efficacy $k_s$ serves as an inverse selection scale, describing the dependence of $f_{MT}$ on $s$. (e) A scatter plot of the fit parameters $f_0$ and $k_s$ for all values of $k$ and $\rho$ reveals that they are effectively not independent parameters, but that $f_0$ entirely determines $k_s$ and vice-versa (for a given population size $N$ and mutation rate $\mu$; here, $N = 10^5$ and $\mu = 0.0005$). The dashed line represents the expect neutral diversity for a circular colony, where $f_0 = \mu R$, with $R = \sqrt{N/\pi}$ the colony radius (*Fusco et al., 2016*). (f) Rescaling all curves by fitted values of $k_s$ and $f_0$, all points fall close to a master curve given by $e^{k_s s}$ (dashed line), motivating the parametrization introduced in (a).

DOI: https://doi.org/10.7554/eLife.44359.028

The following source data and figure supplements are available for figure 5:

**Source data 1.** Source data for *Figure 5*.
DOI: https://doi.org/10.7554/eLife.44359.031

**Figure supplement 1.** Clone size distribution $P(X > x)$ for colonies grown on smooth (a) and rough (b) substrates in simulations.
DOI: https://doi.org/10.7554/eLife.44359.029

**Figure supplement 1—source data 1.** Source data for *Figure 5—figure supplement 1* (Clone size distribution for various $s$ for $\rho = 0$ and $\rho = 0.4$).
DOI: https://doi.org/10.7554/eLife.44359.030

two sides of the same coin: environmental heterogeneity alters the growth pattern of the colony, which in turn affects both the neutral diversity and the selection efficacy.

Our minimal model has shown that simple uncorrelated heterogeneities can have a strong impact on the evolutionary dynamics and morphology of the range expansions. Remarkably, even incomplete obstacles that merely slow down growth can generate large roughness and quasi-neutral population genetics when they are at intermediate density. The reason is that, because the positioning and dynamics at the front are so important in range expansions, a slowdown has similar effects to a complete halt. Regarding environmental heterogeneity as simply another source of (extrinsic) noise,

it is perhaps not surprising from a classical population genetics perspective that the addition of noise effectively weakens selection, as other sources of noise, such as small population sizes, are known to push evolutionary dynamics towards the neutral limit (*Gillespie, 2004*). However, as we show in the following, the environmental heterogeneity in our simulations changes the evolutionary dynamics on a fundamental level that is not consistent with a mere increase in total noise level. Consider the neutral diversity $f_0$ in *Figure 5e*, which corresponds to the rate at which neutral mutations accumulate in the population. On average and in the absence of environmental heterogeneity, this rate is $\mu(N/\pi)^{1/2}$ since a fraction $\mu$ of cells at the population front acquire new neutral mutations in every generation, and the front size scales as the square root of the population size $N$ (*Fusco et al., 2016*). Importantly, this result is independent of the level of noise in the system since it concerns only the average over many populations. Since the population size is the same across all our simulations, we would expect the same neutral diversity for all parameter values $\rho$ and $k$. By contrast, our simulations show clear systematic deviations from the expected value (dotted line in *Figure 5e*).

To further characterize the qualitative changes on the neutral dynamics induced by environmental heterogeneity, we computed the spatially resolved phylogenetic tree of the population, obtained by tracing the lineages of all individuals at the population front back to the origin, focusing on the extreme cases of no ($\rho = 0$) and critical ($\rho = \rho_c$) obstacles (i.e., $k = 0$). For intermediate $\rho$, there is a crossover length set by the obstacle density. On length scales much shorter than this crossover length, the dynamics correspond to that in heterogeneous environments, while the homogeneous dynamics are recovered on length scales much larger than the crossover length (see *Theory* for details and *Figure 4—figure supplement 1b* for an illustration of the crossover length at intermediate $\rho$).

As shown in *Figure 6a,b*, the lineage tree has a vastly different appearance depending on the environmental heterogeneity. Without obstacles (panel a) the lineages are relatively straight and roughly aligned with the radial direction. By contrast, at the critical obstacle density, where the colony has a rough exterior (*Figure 4c*), lineages are erratic and often have segments oriented perpendicular to the radial direction.

To quantify the differences in the lineage structure between the two scenarios, we focused on the pair coalescence 'time' $T_2$, that is the time when two individuals, sampled a distance $\Delta x$ apart, had their last common ancestor (measured in lattice sites, see *Figure 6c*), as well as the strength of lineage fluctuations, that is how much lineages deviate from straight lines over time. The strength of lineage fluctuations determines how likely two randomly lineages are to meet ('coalesce') and thus directly shapes the coalescence structure of the population (*Korolev et al., 2010*; *Chu et al., 2018*).

The lateral lineage fluctuations $l_\perp$ (see Materials and methods for details) scale with distance $t$ from the origin as $l_\perp \sim t^\xi$, where a larger value of $\xi$ indicates rougher boundaries. We find that lineages in rough colonies are not only rougher in absolute value, but also in terms of their scaling in rough colonies. Whereas in the standard Eden model we recover the known (super-diffusive) scaling $\xi = 0.66 \pm 0.006$ (*Kardar et al., 1986*), we find a larger scaling exponent $\xi = 0.86 \pm 0.006$ in rough colonies ($\rho = 0.4$). This is consistent with the corresponding change of the statistical properties of the colony interface, which transitions from the Kardar-Parisi-Zhang (KPZ) universality class to the quenched Edwards-Wilkinson (QEW) universality class (see Materials and methods and *Figure 4—figure supplement 1*). The changes in lineage fluctuations in the presence of environmental heterogeneity are reflected in the shape and orientation of individual neutral clones (*Figure 6—figure supplement 1*). Mutant clones have an approximately ellipsoidal shape oriented preferentially along the radial direction in the absence of heterogeneity, whereas they have essentially random orientations in rough colonies (*Figure 6—figure supplement 1c*), in agreement with the observation that lineages lose their radial orientation as the number of obstacles increased. Similarly, we measured the scaling of clone widths $l_\perp$ with its length $l_\parallel$ as $l_\perp \sim l_\parallel^\zeta$, where the exponent $\zeta$ quantifies the anisotropy of the clones ($\zeta = 0$ corresponding to clones whose width is independent of their length, $\zeta = 1$ corresponding to isotropic clones). In our simulation, $\zeta$ changed from $\zeta = 2/3$ for $\rho = 0$ (consistent with KPZ interface statistics [*Fusco et al., 2016*]) to $\zeta \approx 0.95$ for $\rho = \rho_c$ (*Figure 6—figure supplement 1d*), indicating roughly isotropic neutral clones.

The increased roughness of lineages is also reflected in the number of successful lineages emanating from the initial population founder. We quantify this by computing the pairwise coalescence time $T_2$ (*Figure 6e,f*) over all pairs of cells at the front of the population. We find that, for a given

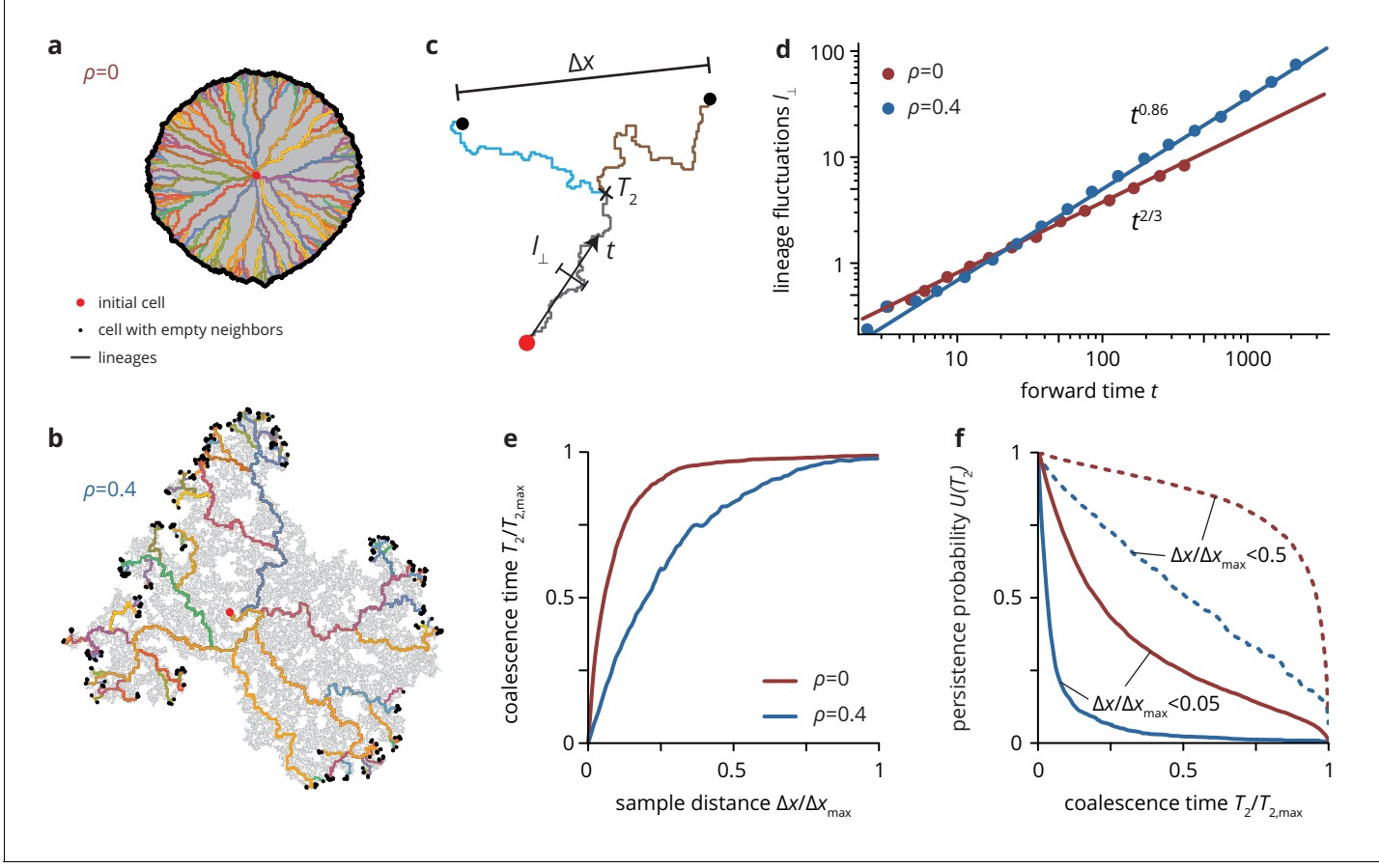

**Figure 6.** Neutral lineage dynamics change with environmental heterogeneity. Lineage trees extracted from simulated colonies without ($\rho = 0$), (**a**) and with critical ($\rho = 0.4$), (**b**) environmental heterogeneity. (**c**) The lineage structure is characterized by the fluctuations $l_\perp$ of the lineages in forward time $t$, and the pair coalescence time $T_2$ (backwards in time) of two samples that are a distance $\Delta x$ apart. (**d**) The lineage fluctuations depend on the environmental heterogeneity: in the absence of environmental disorder, lineage fluctuations scale as $t^{2/3}$, whereas the lineages are rougher in strong disorder, scaling as $t^{0.86}$. Strong lineage fluctuations in heterogeneous environments also impact the mean pair coalescence time of two samples a distance $\Delta x$ apart by allowing lineages from distant regions of the colony to coalesce earlier than without environmental heterogeneity (panel e). As a consequence, the persistence probability $U(T_2)$, that is the probability that two sampled lineages have a pair coalescence time greater than $T_2$ is typically greater in homogeneous than in heterogeneous environments, even when conditioning on small distance between samples (solid lines, $\Delta x < 0.05\Delta x_{max}$; dashed lines, $\Delta x < 0.05\Delta x_{max}$).

DOI: https://doi.org/10.7554/eLife.44359.032

The following source data and figure supplements are available for figure 6:

**Source data 1.** Source data for *Figure 6*.
DOI: https://doi.org/10.7554/eLife.44359.035
**Figure supplement 1.** Environmental heterogeneity alters the shapes of mutant clones.
DOI: https://doi.org/10.7554/eLife.44359.033
**Figure supplement 1—source data 1.** Source data for *Figure 6—figure supplement 1* (Clone angles and anisotropy exponents for neutral mutations for various $\rho$).
DOI: https://doi.org/10.7554/eLife.44359.034

distance $\Delta x$ between the sample pairs, the relative coalescence time and persistence probability (i.e., the probability of not having a common ancestor until time $T_2$, shown in panel f) is always smaller in the presence of obstacles. This indicates that fewer lineages reach the population edge in the presence of environmental heterogeneity. This makes intuitive sense from the phylogenetic trees shown in *Figure 6a,b*, where in rough colonies all individuals at the front coalesce quickly into a small number of large lineages. Thus, for any mutation to be successful and grow into a large clone, it has to belong to one of those large lineages. Since the number of those lineages is small, such mutations

are extraordinarily unlikely; at the same time, if they occur, they can grow to large size simply by virtue of having occurred in a fortuitous location.

## Discussion

Using a combination of bacterial colony experiment and population genetics theory, we examined how environmental heterogeneity can shape the population genetics of range expansions. By growing colonies on heterogeneous substrates, we found that microscale ridges and troughs in the growth substrate were enough to dramatically alter the growth dynamics and morphology of the colonies as well as reduce the ability of beneficial mutations to establish and expand. Time-lapse microscopy and minimal model simulations showed that this reduction in selection efficacy on heterogeneous substrates can be explained by a local pinning of the colony front. Since mutations occur only within the growing population at the front, the properties of the front dictate the evolutionary dynamics, including the strength of selection and the size of individual clones. Local pinning impacts the dynamics at the front by reducing the expansion speed of some parts of the population, leading to an effective reduction in the number of expansion paths that can actively contribute successful mutations. Thus, most individuals and their clones will get stuck in dead-ends, and only a few lucky individuals will be able to find the paths along unpinned front positions and be able to establish a large clone. Once established, though, the size of a mutant clone is roughly independent of its fitness because it is constrained by the network of obstacles. Thus, the evolutionary success of a mutation, that is whether a sector can form or not, and how large a mutant clone can get, depends *entirely* on where it arises and not at all on its fitness. In this sense, locally pinned expansions bear little resemblance to unconstrained radial expansions (see *Figure 7*). Rather, expansions along each available path more closely correspond to linear expansions, for example along a coastline, where mutations spread deterministically after local establishment (*Fisher, 1937*).

While we have focused here on microbial populations, we expect these results to hold more generally. This is one of the conclusions of our deliberately minimal model, which showed that simple uncorrelated heterogeneities are enough to create a strong impact on the population genetics and morphology of the range expansions. These effects persist even when the heterogeneity does not present as impassable obstacles but merely slows down growth. This is because the properties of the front dictate the evolutionary dynamics, since mutations occur only within the growing population at the front, such that a slowdown have comparable effects on the evolutionary dynamics as a

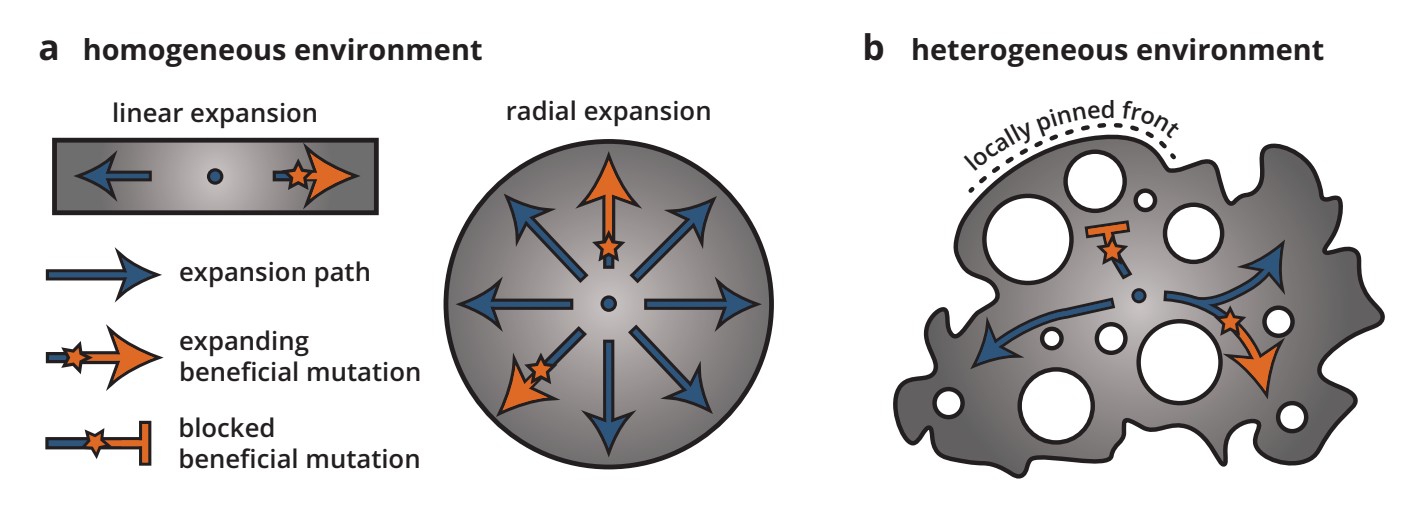

**Figure 7.** Cartoon of how environmental heterogeneity shapes evolutionary dynamics during range expansions. In homogeneous environments (**a**), a locally established beneficial mutation (orange star), having survived genetic drift, can expand freely (arrows). By contrast, in heterogeneous environments (**b**), a beneficial mutation can become trapped in pinned stretches of the population, whereas expansion along open paths resembles a one-dimensional expansion.

DOI: https://doi.org/10.7554/eLife.44359.036

complete halt. Since our model does not include any particular biological details, its results may apply generally in systems with front-limited growth and environmental heterogeneity. Thus, we would expect a reduced selection efficacy to generalize to other dense cellular populations in disordered environments, such as tumors and biofilms, but also to macroscopic range expansions of invasive species (*With, 2002*). When a population undergoes a range expansion, it will arguably not experience a completely homogeneous environments: at the very least, some areas will be more hospitable than others, but other parts of the environment may also be entirely inaccessible to the population because of, for example rivers and lakes, a strong local competitor or predator, or lack of resources. Environmental heterogeneity is thus arguably the rule rather than the exception.

Despite its simplicity, our minimal model reproduces many experimental findings qualitatively, such as a rougher colony morphology and a reduced efficacy of selection in the presence of environmental heterogeneity (in experiments, by a factor of 2.5 - 4 depending on fitting strategy). However, the model cannot quantitatively account for all our experimental results. For instance, the model predicts that the frequency of neutral and deleterious mutants should be greater in heterogeneous than in homogeneous environments (*Figure 5d*). By contrast, in our experiments, we find only about half as many neutral mutants in rough colonies as in smooth colonies and comparable mutant frequencies for deleterious mutations (*Figure 2d*). A potential reason for this discrepancy may be that the random pattern imposed in our experiments is not correlation-free as in the simulations (*Figure 1—figure supplement 3*), which may impact the dynamics of mutant clones as follows. A beneficial mutation has to overcome genetic drift, and to do so, it must grow to a lateral size large enough for selection to take over (*Gralka et al., 2016b*). However, if the characteristic length scale of the environmental heterogeneity is smaller than this 'establishment size', then the evolutionary dynamics is effectively neutral. On the other hand, a deleterious mutation born on a ridge or in a trough never grows to large enough size to 'see' the disorder in the first place and thus its dynamics are largely unaffected by the environmental heterogeneity. In the simulations, however, the heterogeneity can be felt on all length scales, such that it affects mutant clones of all sizes the same way.

We expect environmental heterogeneity to impact not only the fate of beneficial mutations. Since deleterious mutations are typically more numerous than beneficial ones, environmental heterogeneity may also increase the chances of an overall *decrease* in population fitness through the accumulation of deleterious mutations. The accumulation of deleterious mutation, called the expansion load, is already more likely in range expansions than in well-mixed populations (*Peischl et al., 2013*; *Lavrentovich et al., 2016*; *Gralka et al., 2016a*). By altering the efficacy of selection in depressing deleterious mutations and elevating beneficial mutations, heterogeneities in the environment may further facilitate the accumulation of deleterious mutations. Depending on the mutational supply, that is the relative rate and effect of deleterious and beneficial mutations, environmental heterogeneity may not only slow down the process of adaptation but also lead to entirely different long-term evolutionary outcomes. As an example, consider *Figure 8*, where we compute the rate of adaptation in range expansions in various habitats for a given distribution of fitness effects (DFE). We find that the rate of adaptation can transition from positive (adaptation over time) to negative (accumulation of deleterious mutations) depending only on the degree of environmental heterogeneity. Thus, environmental heterogeneity can fundamentally alter the evolutionary dynamics of range expansions.

## Materials and methods

### Experimental methods

#### Strains and growth conditions

We used an *E. coli* MG1655 strain transformed with the plasmid pB10 (*Schlüter et al., 2003*). pB10 is a 65 kB plasmid isolated from sewage sludge that confers resistance to several antibiotic resistance including tetracyclines and has an inserted RFP gene. Hence, cells containing pB10 ('wild type') are red fluorescent and resistant to tetracycline. The plasmid is lost sporadically (*De Gelder et al., 2007*), and the resulting cells ('mutants') are non-fluorescent and susceptible to tetracycline, but display a higher growth rate in the absence of antibiotics (characterized below). We refer to the loss of the plasmid as a 'mutation' of known fitness effect and occurrence rate, both of which we characterize below.

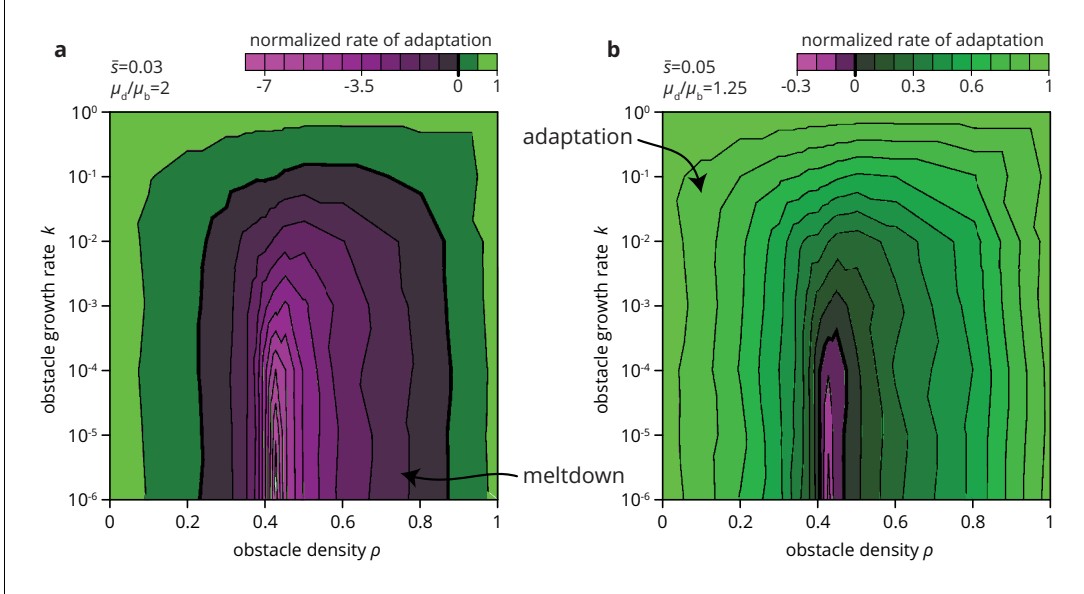

**Figure 8.** Environmental heterogeneity can alter evolutionary outcomes. Rate of adaptation, defined as $\int D(s)sf_{\mathrm{MT}}(s)ds$, computed for an exponential distribution of fitness effects $D(s)$ at various values of the environmental parameters $k$ and $\rho$ (see Materials and methods). The distribution of fitness effects is characterized by two parameters: the ratio $\mu_d/\mu_b$ of deleterious to beneficial mutation rate ($\mu_d/\mu_b = 2$ in panel a, 1.25 in b), and $\bar{s}$, the average fitness effect of a mutation ($\bar{s} = 0.03$ in panel a, 0.05 in b). The rate of adaptation is normalized to its maximal value at $\rho = 0$. Green tones correspond to adaptation (accumulation of beneficial mutations), magenta tones to accumulation of deleterious mutations.

DOI: https://doi.org/10.7554/eLife.44359.037

All experiments were performed in LB at 37 °C in a humidified environment. For solid media, 2% agar was added before autoclaving. Varying concentrations of doxycycline, a tetracycline that displays higher stability in agar plates than tetracycline itself, were added to freshly autoclaved media after cooling to about 60 °C and poured immediately. Plates were dried in the dark for at least 24 hr before use.

### Fitness measurements

We measured the fitness difference $s$ between wild type and mutant cells using the colliding colonies assay (*Korolev et al., 2012*; *Gralka et al., 2016b*). Briefly, a mutant clone was first isolated and then grown independently of the wild type overnight. In the wild type, the plasmid was maintained by adding 10 μg/ml doxycycline to the overnight culture. After growth overnight, cultures were diluted 1:10, grown for about 1.5 hr, and then washed twice in PBS to remove residual doxycycline. 1 μl droplet of each strain were spotted on agar plates about 2 mm apart. After drying, colonies were grown for 3 days and then imaged under the a Zeiss Axiozoom v16 microscope. The resulting images were used to estimate fitness differences by fitting a circle onto the mutant-wild type interface. The results are shown in *Figure 2—figure supplement 1*: without doxycyline, mutants have a 20–25% advantage over the wild type. Both strains have equal growth rate around ≈0.35 μg/ml, and the mutants grow more slowly than the wild type at higher concentration of doxycycline.

For the growth rate measurements in *Figure 2*, colonies were grown from single cells on both rough and smooth plates in a temperature-controlled growth chamber and imaged overnight on a Zeiss Axiozoom v16 microscope. The resulting time lapse movies were binarized and the colony areas extracted.

### Mutation rate experiment

To measure the rate of plasmid loss ('mutation rate'), we grew 48 well-mixed populations from a small number of wild-type cells for about seven generation (i.e., from about 10 to about 1000 cells). The inoculum did not contain any mutants because the culture used to inoculate the populations contained selective amounts of doxycycline. The populations were grown either without doxycycline

or at 1 µg/ml doxycycline, which corresponds to the low and high end of concentrations used in our experiments, respectively. After seven generations, each population was plated and the number of red (WT) and gray (MT) colonies was counted via automated image analysis. The resulting frequencies of mutants were used to infer the mutation rate by computing the maximum likelihood against simulations of the process at different mutation rates and fitness differences, as follows.

## Statistical inference of mutation rate

To estimate the mutation rate, we performed maximum likelihood estimation based on probability density distributions obtained from simulations, as follows: starting from a Poisson distributed number of initial cells, 48 populations go through about seven generations, where every wild type has a chance µ per division to produce a mutant. Mutant cells grow at a growth rate $(1 + s)$ relative to the wild type. We performed 50000 simulations for each value of $s$ and µ and computed the likelihood of each parameter combination $\theta = s$, µ as

$$\ell = \sum_{i=1}^{48} \ln f(x_i | \theta), \tag{1}$$

where $f(x_i)$ is the probability of observing $x_i$ under the simulation model, which we estimated from the simulation histograms. We treat $s$ as a free parameter that we can later compare to the experimentally measured value. The precise value of $s$ does not affect the inferred value of $mu$ very strongly. This is because the number of generations is small in our experiment and a faster-growing mutant can gain at most a factor of four more cells than the wild type. The global maximum likelihood value $\mu^*$ is obtained for $s = 0.3$ and $s = -0.05$ for doxycycline concentrations of 0 µg/ml and 1 µg/ml, in good agreement with our measured values of $s$ (see **Figure 1—figure supplement 1**). The error is estimated from the curvature of the likelihood as $\delta\mu \geq 1/|\partial_\mu^2 \ell|_{\mu^*}$. We find $\mu_{0\mu g/l} = 0.003 \pm 0.00055$ and $\mu_{1\mu g/l} = 0.009 \pm 0.00068$.

## Main experiment and analysis

Our main experiment consists in the growth of colonies from single wild-type cells on agar plates (each containing 20 ml of LB +2% agar) containing varying concentrations of doxycycline. The agar plates were either *smooth*, standard agar plates, or *rough*. Rough plates were created by pouring the agar at a temperature of about 60 °C and then gently lowering filter paper (VWR Grade 410 Filter Paper, Qualitative, pore size between 9 and 20 µm) onto the liquid agar, where it remained until the agar had solidified. The filter paper was then removed from the hard agar surface with tweezers, resulting in a patterned agar surface.

Overnight culture of the wild type grown in LB with 10 µg/ml doxycycline was washed and diluted in PBS to give between 3 and 10 colonies per plate. About $n = 20 - 30$ colonies per condition were analyzed (except for smooth plates without doxycycline, where $n = 8$; see **Table 1**). After 72 hr of growth, the colonies were imaged on a Zeiss Axiozoom v16 microscope and the resulting images binarized to create a mask of the colony. Mutant clones we found manually with ImageJ. The mutant

**Table 1.** Number of colonies assayed for each condition.
Pilot experiment one is reported in **Figure 2—figure supplement 5**, experiment two is reported in the main text.

**Experiment 1 (Figure 2—figure supplement 5)**

| dox (µg/ml) | 0 | 0.2 | 0.4 | 0.6 | 0.8 |
|---|---|---|---|---|---|
| n (smooth) | 13 | 16 | 16 | 20 | 21 |
| n (rough) | 11 | 20 | 16 | 19 | 14 |

**Experiment 2 (main text)**

| dox (µg/ml) | 0 | 0.05 | 0.1 | 0.15 | 0.2 | 0.25 | 0.3 | 0.35 | 0.4 | 0.5 | 0.6 | 0.7 | 0.9 | 1.1 |
|---|---|---|---|---|---|---|---|---|---|---|---|---|---|---|
| n (smooth) | 8 | 30 | 37 | 26 | 31 | 27 | 32 | 41 | 30 | 36 | 30 | 24 | 31 | 28 |
| n (rough) | 20 | 24 | 32 | 20 | 32 | 25 | 19 | 22 | 31 | 28 | 18 | 19 | 27 | 29 |

DOI: https://doi.org/10.7554/eLife.44359.023

frequency per colony was measured as the total mutant area divided by the total area of the colony. The mutant frequency as a function of radius was measured by progressively shrinking the colony mask.

The whole experiment was performed twice. The main text focuses on the second experiment; the results (mutant frequency $f_{\mathrm{MT}}$) from the first experiment are shown in SI *Figure 2—figure supplement 5*.

## Measuring substrate roughness

The properties of the randomly patterned agar surfaces used for the 'rough' condition were assessed using a Dektak 150 profilometer with a 12.5 µm stylus. The run length was 10,000 um, with samples taken over 122 s at 0.273 µm per sample. The measurement range was set to 524 µm. The resulting traces were tilted to have zero slope; roughness was defined as the standard deviation of height values.

## Simulations

To simulate growing bacterial colonies, we employ an Eden model (*Eden, 1961*) on a square lattice that we generalized to include mutations with fitness effect $s$ and environmental disorder. To simulate environmental disorder, we first initialize the lattice with a number of disorder sites at a density $\rho$. Disorder sites are characterized by a reduced growth rate $k$ ($0 \leq k < 1$). For $k = 0$, the disorder sites are impassable (we call this type of disorder sites *obstacles*); equivalent models have also been used to simulate epidemics, where the obstacles represent resistant sites (*Herrmann, 1986*). Without disorder, our model is identical to that used in *Fusco et al. (2016)* and *Gralka et al. (2016b)*, and its interfaces are known to be well described by a standard model for stochastically growing interfaces, the KPZ equation (*Kardar et al., 1986*).

The population is initiated with a single filled site in the center. In each time step, a site with empty neighbor sites is chosen with probability proportional to its growth rate $k \times (1 + s)$ to divide into a randomly chosen empty neighbor site (*Figure 4a*). Upon division, a wild-type site acquires a single mutation with probability $\mu$, potentially conferring a fitness advantage or disadvantage $s$; already mutated sites do not acquire further mutations. The populations are grown until the same number of lattice sites $N$ is filled; for strong environmental noise this results in colonies that are optically larger (see, e.g., *Figure 4c*). For *Figures 5* and *6*, $N = 10^5$.

The mutant frequency $f_{\mathrm{MT}}$ is computed as the number of mutant lattice sites at the end of the simulation, divided by $N$. To arrive at *Figure 5*, we fitted the curves $f_{\mathrm{MT}}(s)$ with an exponential,

$$f_{\mathrm{MT}}(s) = f_0 e^{k_s s}, \tag{2}$$

where we call $f_0$ the neutral diversity and $k_s$ the selection efficacy, which measures the susceptibility of the mutant frequency to changes in $s$.

To derive the phylogenetic trees in *Figure 6*, the locations of mother and daughter cell for each cell division were recorded for 50 colonies, allowing us to trace the lineages of all cells at the front back to the initial cell. From the resulting lineages, we computed the lineage fluctuations $l_\perp$ as the root-mean-squared transverse deviation from the radial direction (*Figure 6d*). The mean pair coalescence time in *Figure 6e* was computed by finding the most recent common ancestor cell for all pairs of cells at the front. The coalescence times $T_2$ (measured backwards in time from the final time point) and the distance $\Delta x$ between the pairs were normalized to be able to compare between the homogeneous and heterogeneous scenarios, then binned and averaged. For *Figure 6f*, we first conditioned on a maximum (normalized) pair distance and then computed the inverse cumulative distribution for two lineages to not have coalesced until time $T_2$.

The rate of adaptation in *Figure 8* was computed by assuming an exponential distribution of fitness effects $D(s)$ for both deleterious and beneficial mutations, that is

$$D(s) = \frac{1}{\bar{s}(1 + R)} \begin{cases} e^{-s/\bar{s}}, & s > 0, \\ e^{-|s|/\bar{s}R}, & s < 0, \end{cases} \tag{3}$$

where $\bar{s}$ and $\bar{s}R$ are the mean fitness effect of beneficial and deleterious mutations, respectively. The

normalization enforces that $R$ corresponds to the total relative frequency of deleterious to beneficial mutations. The rate of adaptation is then computed as

$$\int ds D(s) s f_{\mathrm{MT}}(s), \tag{4}$$

using the values $f_{\mathrm{MT}}(s)$ from *Figure 5*. For the parameters in *Figure 8*, $D(s)$ goes to zero sufficiently quickly such that the limited range in $s$ of the simulations does not impact the integral.

## Theory

Interfaces created by Eden model simulations fall into the KPZ universality class, governed by the KPZ equation for the height $h(x,t)$ (*Family and Vicsek, 1985*; *Kardar et al., 1986*). In one dimension, starting from a line in a simulation box, the colony surface is described by

$$\partial_t h(x,t) = v_\infty + D\partial_x^2 h + \lambda(\partial_x h)^2 + \eta(x,t), \tag{5}$$

where $v_\infty$ is the final speed of front propagation and $\eta(x,t)$ is zero-mean Gaussian random noise $\delta$-correlated in space and time describing the noise associated with the growth process. This equation generates a set of characteristic exponents that govern the roughness of the colony front and of sector boundaries. In particular, the surface height is described in terms of its root mean squared fluctuations around the mean by a Family-Viscek scaling relation (*Vicsek and Family, 1984*)

$$h(t) \propto L^\alpha \mathcal{F}\left(t/L^{\alpha/\beta}\right), \tag{6}$$

where

$$\mathcal{F}(x) = \begin{cases} x^\beta, & x \ll 1, \\ 1, & x \gg 1. \end{cases} \tag{7}$$

The KPZ universality class is characterized by the roughness exponent $\alpha = 1/2$ and the temporal exponent $\beta = 1/3$; if $\lambda = 0$, the resulting universality class is called the Edwards-Wilkinson (EW) universality class characterized by $\alpha = 1/2$ and $\beta = 1/4$. The ratio $z = \alpha/\beta$ is sometimes called the dynamical exponents. It relates the size of lateral fluctuations $l_\perp$ to the time $t$ as

$$l_\perp \sim t^{1/z}. \tag{8}$$

This relationship explains the fluctuations in sector boundaries in *E. coli* colonies and Eden model simulations, and can be used to derive exponents for the site frequency spectrum and establishment probabilities in Eden model colonies (*Hallatschek et al., 2007*; *Fusco et al., 2016*; *Gralka et al., 2016b*). In the simulation presented here, the scenario without environmental disorder is described by *Equation (5)*, explaining the site-frequency spectrum and the anisotropy exponents $\zeta = 2/3 = 1/z$ and the lineage fluctuations scaling in *Figure 6*.

The effect of environmental *quenched* disorder on the kinetic roughening of interfaces has been investigated in a range of experiments (see *Barabási and Stanley, 1995* for a review). Exponents obtained from experiments are in the range of $\alpha \approx 0.6 \ldots 0.9$. To model driven interface growth in disordered media, quenched environmental disorder can be included by considering a noise term $\zeta(x, h(x,t))$ that does not explicitly depend on time (*Kessler et al., 1991*).

$$\partial_t h(x,t) = F + D\partial_x^2 h + \lambda(\partial_x h)^2 + \zeta(x, h(x,t)). \tag{9}$$

Here, $F$ is driving 'force' fulfilling the same role as $v_\infty$ in *Equation (5)*. Since the noise explicitly depends on the interface position, $F$ cannot be transformed away and thus emerges as a new parameter that can be thought of as a force pushing the interface through the disordered media. An important consequence of quenched noise is the emergence of a critical force $F_c$ below which the interface becomes pinned (*Tang and Leschhorn, 1992*). For $F > F_c$, a depinning transition takes place that is well characterized in 1 + 1 dimensions (*Amaral et al., 1995*). For $F \to F_c^+$, large regions of the interface of size $\xi \sim |F_c - F|^{-\nu}$ are pinned, and the front speed increases as $|F - F_c|^\theta$ (see *Table 2*).

**Table 2.** Characteristic exponents for the KPZ and QEW universality classes and measured in our simulations, which are in good agreement with the literature values in *Amaral et al. (1995)*.

| Exponent | KPZ | This study ($\rho = 0$) | qEW | This study ($\rho = \rho_c$) |
|---|---|---|---|---|
| $\alpha_{\mathrm{loc}}$ | 1/2 | 0.5 ± 0.05 | 0.92 ± 0.04 | 0.9 ± 0.05 |
| $\alpha_{\mathrm{G}}$ | 1/2 | 0.5 ± 0.05 | 1.23 ± 0.04 | 1.15 ± 0.05 |
| $\beta$ | 1/3 | 0.3 ± 0.03 | 0.86 ± 0.03 | 0.78 ± 0.05 |
| $\theta$ | N/A | N/A | 0.24 ± 0.03 | 0.21 ± 0.03 |
| $\zeta$ | 2/3 | 0.66 ± 0.02 | - | 0.86 ± 0.04 |

DOI: https://doi.org/10.7554/eLife.44359.038

Simulations and numerical integrations of *Equation (9)* have characterized the pinned and moving phases and uncovered two universality classes as $F \to F_c^+$ (*Amaral et al., 1995*): if $\lambda$ diverges near the depinning transition, one speaks of the (quenched) QKPZ universality class; its exponents $\alpha = \beta \approx 0.633$ in the pinned phase are understood analytically through an analogy with the directed percolation class, whereas in the moving phase $\alpha \approx \beta \approx 0.75$. If $\lambda \to 0$ instead, one speaks of the quenched Edwards-Wilkinson (QEW) universality class with $\alpha \approx 0.92$ and $\beta \approx 0.82$ in the moving phase (see also *Table 2*); a functional renormalization group calculation gives $\alpha = 1$ and $\nu = 1/(2 - \alpha)$ in one dimension (*Nattermann et al., 1992*). For $F > F_c$, there is a transition from the QKPZ/QEW universality class to the appropriate universality class with annealed noise.

Our generalized Eden model simulations with obstacles exhibits the same pinning transition for an obstacle density $\rho \approx 0.4$. At the transition, the resulting colonies are (site) percolation clusters on the square lattice, from whose interfaces we measure exponents that are in excellent agreement with the QEW universality class (see *Table 2* and *Figure 4—figure supplement 1*). This is consistent with the finding of *Moglia et al. (2016)*, who used a slightly more complex simulation algorithm to model the growth of cancer cell monolayers. Without obstacles, our simulations reproduce earlier findings (*Family and Vicsek, 1985*) (*Table 2*). In particular, we find $z = 3/2$ without obstacles and $z \approx 1.15$ at the critical obstacles density, which allows us to compute the scaling exponent of the sector boundaries from *Equation (8)* as $\zeta = 1/z$. This gives $\zeta = 2/3$ and $\zeta \approx 0.86$, in excellent agreement with the lineage fluctuation exponents $\zeta = 0.66$ and $\zeta \approx 0.86$ in *Figure 6d*. The clone size distribution $P(A > a)$ for bubbles is predicted to scale as $a^{-1/(1+z)}$, which gives $a^{-2/5}$ in the KPZ case and predicts $a^{-0.46}$ for QEW. In our simulation, we indeed observe similar clone size scaling without obstacles and at the critical obstacle density (see *Figure 5—figure supplement 1b*, inset).

For a given obstacle density $\rho$ the interface displays QEW scaling on length scales much shorter than the crossover scale $\ell(\rho)$ and KPZ scaling on much larger length scales, see *Figure 4—figure supplement 1b*. We expect this crossover to shape the population genetics for intermediate $\rho$. Consider first neutral mutations, whose clone sizes we quantify in *Figure 5—figure supplement 1*. Clones with a width $\ell_\perp \ll \ell(\rho)$ will exhibit the QEW scaling, whereas clones with a width $\ell_\perp \gg \ell(\rho)$ will have the standard KPZ scaling. Since large clones (bubbles and sectors) dominate the mutant fraction $f_{\mathrm{MT}}$, the neutral dynamics is thus dominated by KPZ scaling if the colony is much larger than $\ell(\rho)$. The situation is less clear for beneficial mutations. Selection induces another length scales $\ell_s$, as we have recently shown (*Gralka et al., 2016b*). The length scale $\ell_s$ is the characteristic width of a beneficial clone below which it behaves neutrally; once the clone width has reached $\ell_s$, selection can overcome drift and drive the clone to fixation. For KPZ interfaces, $\ell_s \sim 1/s$. Based on our earlier results, we can conjecture that for large systems ($\gg \ell(\rho)$), the selection length $\ell_s$ can become larger than $\ell(\rho)$ if $s$ is sufficiently small. Thus, for $s \ll 1/\ell(\rho)$, we expect to recover the smooth ($\rho = 0$) behavior, with renormalized coefficients.

## Acknowledgements

The strain pB10 was a generous gift by Thibault Stalder and Eva Top. Surface profilometry was performed with the help of Kurt Broderick. We would like to thank Jayson Paulose, Diana Fusco, and Jona Kayser, and all members of the Hallatschek lab for helpful discussions. We also thank Richard Neher, Maxim Lavrentovich, and an anonymous reviewer for their insightful and helpful comments.

Research reported in this publication was supported by the National Institute of General Medical Sciences of the National Institutes of Health under award R01GM115851, a National Science Foundation CAREER Award (#1555330) and a Simons Investigator award from the Simons Foundation (#327934).

## Additional information

### Funding

| Funder | Author |
| --- | --- |
| Simons Foundation | Oskar Hallatschek |
| National Science Foundation | Oskar Hallatschek |
| National Institute of General Medical Sciences | Oskar Hallatschek |

The funders had no role in study design, data collection and interpretation, or the decision to submit the work for publication.

### Author contributions

Matti Gralka, Conceptualization, Data curation, Software, Formal analysis, Validation, Investigation, Visualization, Methodology, Writing—original draft, Writing—review and editing; Oskar Hallatschek, Conceptualization, Supervision, Funding acquisition, Methodology, Project administration, Writing—review and editing

### Author ORCIDs

Matti Gralka https://orcid.org/0000-0003-4599-1859
Oskar Hallatschek https://orcid.org/0000-0002-1312-5975

### Decision letter and Author response

Decision letter https://doi.org/10.7554/eLife.44359.043
Author response https://doi.org/10.7554/eLife.44359.044

## Additional files

### Supplementary files

• Transparent reporting form
DOI: https://doi.org/10.7554/eLife.44359.039

### Data availability

All data generated or analysed during this study are available at Mendeley. Source data files have been provided for all figures.

The following dataset was generated:

| Author(s) | Year | Dataset title | Dataset URL | Database and Identifier |
| --- | --- | --- | --- | --- |
| Matti Gralka | 2019 | Code & Data for: Environmental heterogeneity can tip the population genetics of range expansions | http://dx.doi.org/10.17632/2fkhp73bc6.1 | Mendeley, 10.17632/2fkhp73bc6.1 |

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
