## [Decision Letter]

Thank you for submitting your article "Environmental heterogeneity can tip the population genetics of range expansions" for consideration by *eLife*. Your article has been reviewed by three peer reviewers, including Richard A Neher as the Reviewing Editor and Reviewer #1, and the evaluation has been overseen by Ian Baldwin as the Senior Editor. The following individual involved in review of your submission has agreed to reveal their identity: Maxim Lavrentovich (Reviewer #3).

The reviewers have discussed the reviews with one another and the Reviewing Editor has drafted this decision to help you prepare a revised submission.

Summary:

Gralka and Hallatschek present an interesting and elegant analysis of the effect of environmental heterogeneity on the efficacy with which populations can select for beneficial mutations and against deleterious mutations. They compare colony growth experiments on agar plates that are either roughened with filter paper or are smooth. The bacterial strains carry a plasmid that is lost at a high rate and loss of this plasmid has a tunable effect on growth. The plasmid is beneficial in presence of doxycycline but induces a cost in absence of drug. The authors show that selection for/against plasmid loss differs dramatically on rough or smooth surfaces. On rough surfaces, populations cannot select efficiently for beneficial mutations, and the fitness effect has little consequences for evolutionary dynamics. These results are then compared to a simple theoretical model of evolution in a heterogeneous environment, which revealed that evolutionary dynamics changes qualitatively as the obstacle density approaches the percolation threshold.

This work with its combination of experiment and theory addresses an important problem in evolutionary biology. However, to make the conclusions more compelling, we would like the authors to address the points below.

Essential revisions:

In their reports and the ensuing discussion, the reviewers agreed that there are two central points that need to be addressed:

1) Experiment and theory are not sufficiently linked. There is no discussion whether the experimental system is plausibly close to a percolation threshold and the nature of the surface roughness remains elusive. The authors should characterize the patterned surface and possibly repeat the experiments with different degrees of roughness. Quantitative measurements of correlations, colony surface roughness, and other length scales in the experiment should be used to explicitly link to theory and make the case that experimental system is close to a percolation threshold. The reduction in growth rate and the difference in overall colony expansion rate should be explicitly measured at different drug concentrations and used to compare with model predictions. In its current form, the experiments motivate the theory, but the theory is not used to quantitatively explain the experimental results.

2) Mutation frequencies *f*_MT_ depend on time and comparing *f*_MT_ in two conditions with different growth rates at a fixed time is not very meaningful. It would be much better to compare trajectories and log-derivatives of *f*_MT_. Some discussion on whether *f*_MT_ should be measured in bulk or on the perimeter would also be welcome.

Please also note that simulation and analysis code should be made available (for example on github) along with suitable documentation.

---

## [Author Response]

Essential revisions:In their reports and the ensuing discussion, the reviewers agreed that there are two central points that need to be addressed:1) Experiment and theory are not sufficiently linked. There is no discussion whether the experimental system is plausibly close to a percolation threshold and the nature of the surface roughness remains elusive. The authors should characterize the patterned surface and possibly repeat the experiments with different degrees of roughness. Quantitative measurements of correlations, colony surface roughness, and other length scales in the experiment should be used to explicitly link to theory and make the case that experimental system is close to a percolation threshold. The reduction in growth rate and the difference in overall colony expansion rate should be explicitly measured at different drug concentrations and used to compare with model predictions. In its current form, the experiments motivate the theory, but the theory is not used to quantitatively explain the experimental results.

We have rewritten the manuscript extensively, added new figures and performed additional experiments following the reviewers’ suggestions to connect theory and experiment:

Firstly, we characterized the “rough” surfaces in detail using profilometry. A representative image of a rough surface is shown in Figure 1—figure supplement 2, where light areas correspond to elevation and dark areas corresponds to depressions, showing that the surface is characterized by elongated ridges and “channels” (Figure 1—figure supplement 2A). Profile traces of the surface allow us to accurately quantify the roughness of the surface (SD 10um, range 50um). In addition, adaptive binarization of the microscopic image allows us to estimate the density of elevations, which may serve as obstacles, as between 0.3 and 0.5 (Figure 1—figure supplement 2D). Although this estimate is very approximate, it nevertheless suggests that the rough surface is rugged in a way that makes it plausible that the colony may be near a percolation transition (but, as we discuss below, a strict percolation transition is not needed for a strong effect on population genetics).

The width of the “channels” in the substrate is robust with respect to the choice of binarization parameters; the channels and ridges are typically about 15-30um wide (Figure 1—figure supplement 2E). This length scale is consistent with the width at which the branch-like outgrowths of the colonies become visible (about 20um). It also coincides with the width of the growth layer, which is the length scale governing the range of mechanical interactions within the colony. This explains why the colony morphology is impacted so strongly by the substrate surface roughness.

We now also comment in manuscript on the fact that dramatic effects can occur in models that do not have a percolation transition, which is one of the conclusions of our minimal model. This is because the properties of the front dictate the evolutionary dynamics, since mutations occur only within the growing population at the front, such that a slowdown have comparable effects on the evolutionary dynamics as a complete halt. In our experiments, we have no real obstacles because growth continues outside of the channels, and thus we have no percolation transition in the experiment – but just like in our theory, where we see a strong response in population dynamics and genetics even for k>0.

Secondly, we now explicitly show the radial growth rate in Figure 2 for both scenarios, which shows that the radial growth rate is lower on rough substrates, but in a way that preserves the relative growth rate differences between mutants and wild types.

Thirdly, we now more clearly specify the role and purpose of theory in this paper as answering the following question: given the obvious complexities of the experimental system (mechanical forces, potential for cell-cell ordering, strong correlations in the heterogeneities of growth substrate) do our key findings hinge on these complexities, or can much simpler uncorrelated heterogeneities in growth rates also have comparable effects on population genetics? Our simple lattice model that is agnostic to biological details clearly confirms that the latter is the case. We discuss the limitations of this approach in this Discussion section:

“Despite its simplicity, our minimal model reproduces many experimental findings qualitatively, such as a rougher colony morphology and a reduced efficacy of selection in the presence of environmental heterogeneity. […] In the simulations, however, the heterogeneity can be felt on all length scales, such that it affects mutant clones of all sizes the same way.”

2) Mutation frequencies f_MT_ depend on time and comparing f_MT_ in two conditions with different growth rates at a fixed time is not very meaningful. It would be much better to compare trajectories and log-derivatives of f_MT_. Some discussion on whether f_MT_ should be measured in bulk or on the perimeter would also be welcome.

Measuring *f*_MT_ on the final colony sizes has the advantage of averaging over a large number of mutant clones, which reduces the noise due to the stochasticity of when a successful mutation arises. In addition, it is simple to measure and interpret, and gives a direct impression of the strong differences in the population genetics between smooth and rough substrates. However, it is accurate that *f*_MT_ depends on time; in principle, if the colony growth rates were extremely different, *f*_MT_ could differ simply because one scenario gives rise to much smaller colonies with corresponding smaller *f*_MT_. In our experiments, final radii were comparable between the two treatments, so this issue does not arise.

As suggested, we have analyzed the mutant frequency over time (i.e., as a function of radius) for all different antibiotic concentrations. The results are presented in Figure 2—figure supplement 2. In both scenarios (smooth/rough substrate), the mutant frequency increases super-linearly with radius; square-root transformed *f*_MT_ values give good linear fits (although noisy for rough substrates), whose slopes show the same tendencies as the original *f*_MT_ plot in Figure 2: on smooth substrate, the slope increases with s for s>0, whereas it stay roughly constant on rough substrates. However, there is no established theory that would predict the quadratic relationship between *f*_MT_ and *r*, and since it is beyond the scope of our paper to develop such a theory, we prefer to have the original figure (*f*_MT_ at the final radius) in the main text, and refer the reader to the new figure supplement.

Finally, we have added a brief discussion on whether *f*_MT_ should be measured in bulk or on the perimeter. In our view, there are several reason that make the use of a bulk measurement (either as a function of radius, or simply on the whole-colony level) preferable over measuring the frequency at the perimeter.

1) For strongly beneficial mutations, the mutant frequency at the perimeter can quickly reach 1, making beneficial mutations more difficult to distinguish in terms of their frequency at the perimeter (although differences remain visible on the colony scale).

2) At the same time, bulk measurements give a conservative, but proportional estimate of the frequency at the perimeter.

3) One may argue that the subpopulation at the perimeter is the only important part in terms of the future of the population. However, especially for dense cellular populations such as biofilms or tumors, we believe it is unclear whether this is true in general, since the answer hinges on the mode of dispersal. Biofilms might form spores or other propagules preferentially in the bulk of the population, and tumor cells undergoing the EMT need not be situated at the tumor edge.

4) Microscopically, clones at the perimeter are difficult to detect reliably because the overall fluorescence intensity is lower.

5) By restricting the analysis to the perimeter, the amount of data available for analysis is lower, yielding noisier data (even if clone detection was perfect at the perimeter).

Our decision to focus on the bulk mutant frequency is now discussed in the main text as follows: “The primary readout of our experiments is the final mutant frequency *f*_MT_ in the colony […]However, this measure can quickly become uninformative for beneficial mutations as mutants overtake the whole perimeter and is often difficult to measure accurately, while the mutant frequency averaged over the whole population gives a conservative, but proportional measure of the mutant frequency at the perimeter.”

Please also note that simulation and analysis code should be made available (for example on github) along with suitable documentation.

We have made all data available under the URL http://dx.doi.org/10.17632/2fkhp73bc6.1 including analysis scripts (Mathematica), simulation code (C++), simulation outputs (txt), as well as raw and edited microscopy images.